# Measuring NDC80 binding reveals the molecular basis of tension-dependent kinetochore-microtubule attachments

Tae Yeon Yoo[1,2]*, Jeong-Mo Choi[3,4], William Conway[5], Che-Hang Yu[6], Rohit V Pappu[3,4], Daniel J Needleman[1,2,6]

[1]Department of Molecular and Cellular Biology, Harvard University, Cambridge, United States; [2]Faculty of Arts and Sciences Center for Systems Biology, Harvard University, Cambridge, United States; [3]Department of Biomedical Engineering, Washington University in St. Louis, St. Louis, United States; [4]Center for Biological Systems Engineering, Washington University in St Louis, St Louis, United States; [5]Department of Physics, Harvard University, Cambridge, United States; [6]John A. Paulson School of Engineering and Applied Sciences, Harvard University, Cambridge, United States

**Abstract** Proper kinetochore-microtubule attachments, mediated by the NDC80 complex, are required for error-free chromosome segregation. Erroneous attachments are corrected by the tension dependence of kinetochore-microtubule interactions. Here, we present a method, based on fluorescence lifetime imaging microscopy and Förster resonance energy transfer, to quantitatively measure the fraction of NDC80 complexes bound to microtubules at individual kinetochores in living human cells. We found that NDC80 binding is modulated in a chromosome autonomous fashion over prometaphase and metaphase, and is predominantly regulated by centromere tension. We show that this tension dependency requires phosphorylation of the N-terminal tail of Hec1, a component of the NDC80 complex, and the proper localization of Aurora B kinase, which modulates NDC80 binding. Our results lead to a mathematical model of the molecular basis of tension-dependent NDC80 binding to kinetochore microtubules in vivo.
DOI: https://doi.org/10.7554/eLife.36392.001

*For correspondence:
taeyeon_yoo@hms.harvard.edu

**Competing interests:** The authors declare that no competing interests exist.

## Introduction

Chromosome segregation errors lead to aneuploidy and micronuclei formation, which are closely associated with cancer, infertility, and birth defects (*Santaguida and Amon, 2015*). Accurate chromosome segregation is believed to result from a process that actively suppresses potential errors. The mechanism of error correction remains unclear, but extensive evidence suggests that it is based on the regulation of the attachment of microtubules to chromosome via the kinetochore, a protein complex assembled at centromeres (*Godek et al., 2015*). Previous works suggested that error correction is largely due to the detachment of kinetochore microtubules (kMTs) being regulated by the tension across centromeres, which selectively destabilizes erroneous kMT attachments bearing low tension and stabilizes proper attachments under high tension (*Nicklas and Ward, 1994*; *Liu et al., 2009*; *Akiyoshi et al., 2010*; *Lampson and Cheeseman, 2011*; *Godek et al., 2015*). However, the molecular mechanism of the tension-dependent regulation of kMT attachments is still poorly understood.

The highly conserved NDC80 complex is the major coupler of the kinetochore to microtubules (*Cheeseman et al., 2006*; *DeLuca et al., 2006*). In human mitotic cells, ~240 NDC80 complexes are recruited at the outer layer of each kinetochore (*Suzuki et al., 2015*) and interact with ~20 kMTs by

**eLife digest** When a cell divides, each new cell that forms needs to contain a complete set of DNA, which is stored in structures called chromosomes. So first, the chromosomes duplicate, and the two copies are held together. A protein structure known as a kinetochore then forms on each copy of the chromosome. The kinetochores act as a pair of hands that pull the chromosome copies apart and toward opposite sides of the dividing cell. They do this by grabbing protein 'ropes' called microtubules that extend toward the chromosomes from each side of the cell.

Kinetochores grip the microtubule ropes more tightly when the connection is under greater tension. This helps the kinetochores to remain attached to the microtubules that will separate the chromosome copies while releasing the microtubules that would pull both copies to the same side. Previous research has shown that hundreds of finger-like structures made out of a protein group called NDC80 extend from each kinetochore 'hand' and attach to the microtubules. What remains a mystery is whether and how the NDC80 fingers grip the microtubules more tightly when tension is greater in cells.

Yoo et al. developed a technique for counting how many of the available NDC80 fingers of a single kinetochore are attached to microtubules within a living human cell. The new technique combines genetic engineering, fluorescence imaging and statistical methods to quantify the attachment of NDC80 to microtubules over time and space.

Yoo et al. found that more NDC80 bound to microtubules when there was greater tension. This relationship between binding and tension depends on an enzyme called Aurora B, which modifies the tip of each NDC80 finger and consequently changes the binding of NDC80 to microtubules. Yoo et al. further showed that Aurora B needs to be properly placed between two kinetochore hands to make NDC80-microtubule binding dependent on tension. Without this tension dependency, chromosomes could segregate unevenly into the newly formed cells – a problem that can lead to cancer, infertility and birth defects. The results presented by Yoo et al. therefore expand our understanding of how these diseases originate and may eventually help researchers to develop new treatments for them.

DOI: https://doi.org/10.7554/eLife.36392.002

directly binding to them (*Cheeseman and Desai, 2008*; *Maiato et al., 2004*; *Rieder, 1982*). In vitro experiments showed that the binding affinity of NDC80 for microtubules decreases upon the phosphorylation of the N-terminal tail of Ndc80/Hec1 protein by Aurora B kinase (*Cheeseman et al., 2006*; *Zaytsev et al., 2014*, *2015*), which may explain the contribution of Aurora B to error correction (*Tanaka et al., 2002*). It is unclear how the biochemical activities of NDC80 and Aurora B result in tension-dependent kMT detachment. The lack of techniques to measure the binding of the NDC80 to kMTs in vivo has been a major obstacle to investigate this.

## Results

### FLIM-FRET measures the fraction of donor-labeled NDC80 complexes engaged in FRET with acceptor-labeled microtubules

Inspired by previous work (*Posch et al., 2010*), we sought to develop a Förster Resonance Energy Transfer (FRET) based approach to directly measure the association between the NDC80 complex and kinetochore microtubules (kMTs) in living cells. We engineered U2OS cells stably expressing Nuf2, a subunit of the NDC80 complex, N-terminally labeled with a cyan fluorescent protein, mTurquoise2 (*Figure 1A*). In this same cell line, we also inserted a tetracysteine (TC) motif at the C-terminus of β-tubulin (TUBB) using CRISPR-induced homologous recombination, which becomes fluorescent after binding to the membrane-permeable dye FlAsH (*Hoffmann et al., 2005*) (*Figure 1A and B*). The small size (six amino acids) of the TC motif minimizes the negative effects of labeling the C-terminus of tubulin, allowing the engineered cells to successfully pass through mitosis (*Andresen et al., 2004*). CRISPR-mediated endogenous tubulin tagging ensures low cell-to-cell variation and a high fraction of labeled β-tubulin, which was estimated to be $26.1 \pm 5.4\%$ (SD) (*Figure 1—figure supplement 1* and see Supplemental experiments in Materials and methods).

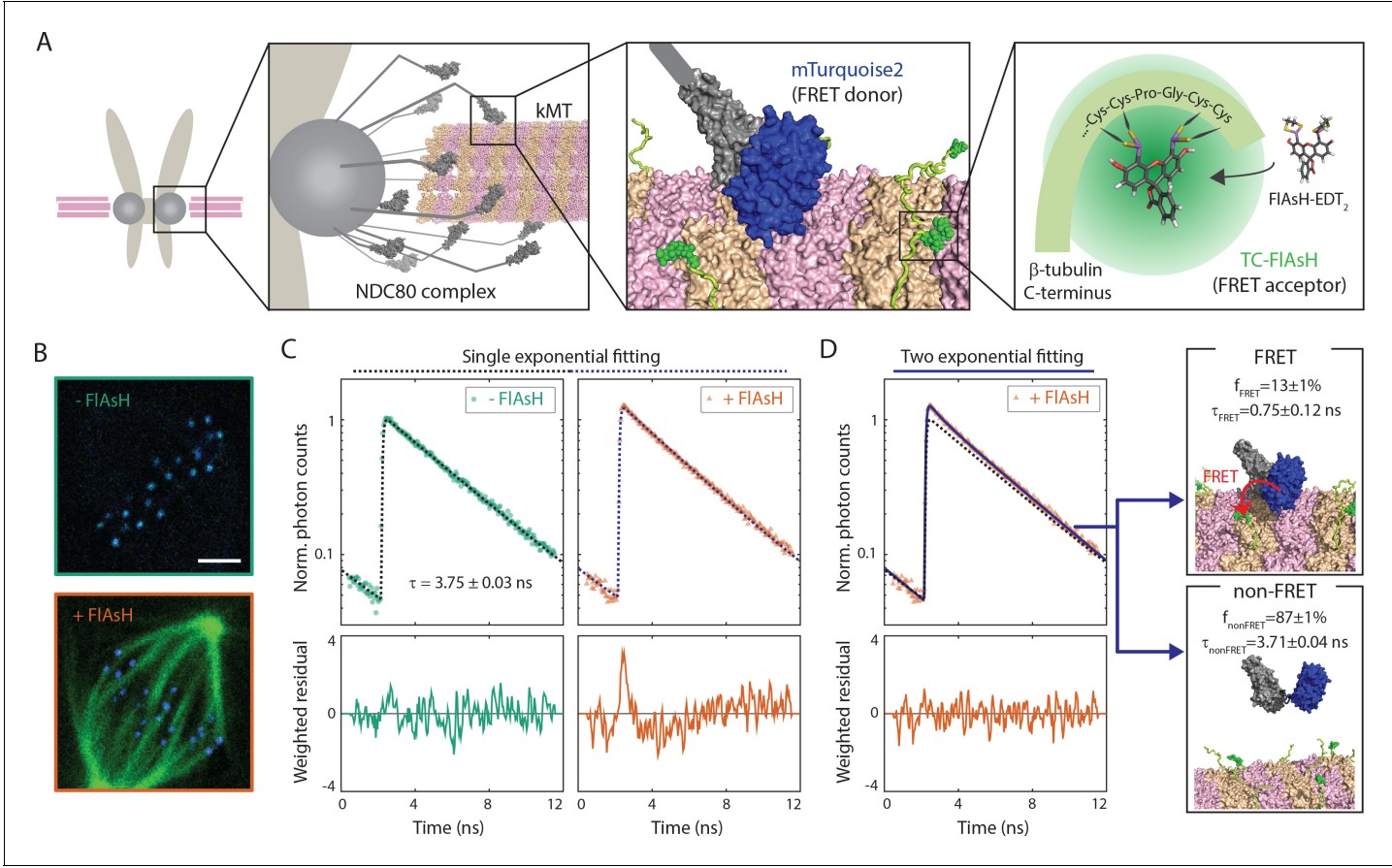

**Figure 1.** FLIM-FRET measurement of NDC80-kMT binding in human tissue culture cells. (**A**) Engineered U2OS cell expressing mTurquoise2-NDC80 and β-tubulin-TC-FlAsH. NDC80 (gray), mTurquoise2 (blue) and TC-FlAsH (green). (**B**) Two-photon microscopy images of the engineered U2OS cells not exposed to FlAsH (top) and exposed to FlAsH (bottom). 3 μm scale bar. mTurquoise2 (blue) and FlAsH (green). (**C**) Example fluorescence decay curves of mTurquoise2-NDC80 in the engineered U2OS cells not exposed to FlAsH (left, green circle) and exposed to FlAsH (right, orange triangle), plotted with the best-fit single-exponential decay models (black and blue dotted lines). Corresponding weighted residuals (the deviation of data from model, divided by the square root of the number of photons) are plotted below after being smoothened to display systematic deviations. (**D**) The fluorescence decay curve of mTurquoise2-NDC80 in the presence of FlAsH labeling (orange triangle, same as (**C**)), plotted with the best-fit two-exponential model (blue solid line). The single-exponential model fit to the fluorescence decay curve in the absence of FlAsH labeling (black dotted line) plotted together for comparison. Corresponding smoothened weighted residual (described above) for the two-exponential model is plotted below. Long- and short-lifetime exponentials correspond to the mTurquoise2-NDC80 populations in non-FRET state and FRET state, respectively, and their relative amplitudes give the fraction of each population. To facilitate the comparison, the fluorescence decay curves in the absence and presence of FlAsH labeling were normalized such that they asymptotically overlap. Data points and source FLIM data are available in Figure 1-Data (*Yoo et al., 2018*).

DOI: https://doi.org/10.7554/eLife.36392.003

The following figure supplements are available for figure 1:

**Figure supplement 1.** β-tubulin labeling fraction measurement.
DOI: https://doi.org/10.7554/eLife.36392.004

**Figure supplement 2.** Kinetochore FLIM-FRET measurement.
DOI: https://doi.org/10.7554/eLife.36392.005

**Figure supplement 3.** Negative control data for NDC80-kMT FLIM-FRET measurements.
DOI: https://doi.org/10.7554/eLife.36392.006

**Figure supplement 4.** Förster radius estimation by FLIM-FRET measurements and Monte Carlo protein simulations.
DOI: https://doi.org/10.7554/eLife.36392.007

**Figure supplement 5.** Characterization and calibration of NDC80-kMT FLIM-FRET measurement.
DOI: https://doi.org/10.7554/eLife.36392.008

We used time-correlated single photon counting (TCSPC) fluorescence lifetime imaging microscopy (FLIM) to quantitatively measure FRET between mTurquoise2 and TC-FlAsH in tissue culture cells (*Figure 1—figure supplement 2*). TCSPC FLIM-FRET provides fluorescence decay curves of the donor fluorophore at each pixel location. If a donor fluorophore has a single-exponential fluorescence decay curve when not engaged in FRET, then when it is engaged in FRET the fluorescence decay curve will also be single-exponential, but with a shorter lifetime. A pixel containing a mixture of such donor fluorophores engaged in FRET and not engaged in FRET displays a fluorescence decay curve that is a sum of two exponentials. Bayesian analysis of the fluorescence decay curves provides a bias-free measurement of the relative fraction of the two exponentials, and hence the fraction of donor fluorophores engaged in FRET (*Yoo and Needleman, 2016*; *Kaye et al., 2017*). In contrast to intensity-based FRET methods, FLIM-FRET is capable of quantifying the fraction of donor fluorophores engaged in FRET when donors and acceptors are differentially distributed in cells, and it is less prone to errors arising from instrumental artefacts and photobleaching (*Berezin and Achilefu, 2010*).

We first characterized the fluorescence decay of mTurquoise2-NDC80 in the absence of FRET by performing FLIM measurement on the engineered U2OS cells (mTurquoise2-NDC80/β-tubulin-TC) that were not exposed to FlAsH (*Figure 1B*, top). We found that their fluorescence decays are well described as a single exponential with a lifetime of 3.75 ± 0.09 ns (SD) (*Figure 1C*, left, and *Figure 1—figure supplement 3A*). As discussed above, this single exponential decay profile is expected when the donor fluorophores do not engage in FRET. We next measured the fluorescence decay of the mTurquoise2-NDC80 in the presence of FlAsH labeling of microtubules. In this case, a single exponential provided a poor fit to the data, exhibiting significant systematic deviations (*Figure 1C*, right). The fluorescence decay in the presence of FlAsH labeling was well fit by a sum of two exponentials with lifetimes 3.71 ± 0.04 ns (SE) and 0.75 ± 0.12 ns (SE) (*Figure 1D*). The long life-time of the two-exponential fit was indistinguishable from the lifetime in the absence of FRET (p=0.68, two-sided Z-test), and thus corresponds to the non-FRET donor population. Therefore, the short-lifetime species is the FRET donor population. The relative amplitude of the short- and long-lifetime exponentials are 0.13 ± 0.01 (SE) and 0.87 ± 0.01 (SE), respectively, thus 13 ± 1% (SE) of donor fluorophores are engaged in FRET.

## FRET between mTurquoise2-NDC80 and FlAsH results from the NDC80-kMT binding

Having demonstrated our ability to measure FRET between mTurquoise2-NDC80 and FlAsH in tissue culture cells, we explored if the FRET is due to the binding of NDC80 to kMTs. We first engineered an alternative construct with mTurquoise2 conjugated to the distally located C-terminus of Nuf2, far removed from kMTs. This alternative construct displayed only a single long-lifetime state in either the presence or absence of TC-FlAsH, arguing that FRET does not result from non-specific interactions (*Figure 1—figure supplement 3B,C and E*). Incubating cells with nocodazole to depolymerize microtubules caused a reduction (p<10$^{-10}$, two-sided Z-test) of NDC80 FRET fraction from 13 ± 1% (SE) to 3 ± 1% (SE) (*Figure 1—figure supplement 3D*). Thus, FRET strongly depends on the presence of microtubules.

We next investigated if NDC80 that is close to kMTs, but not bound to them, can lead to appreciable FRET. Answering this requires knowing the Förster radius between mTurquoise2 and TC-FlAsH, which we determined to be 5.90 ± 0.10 nm (SE) through a combination of FLIM measurements and Monte Carlo simulations (see *Figure 1—figure supplement 4* and Supplemental experiments in Materials and methods). We next performed large-scale Monte Carlo simulations of mTurquoise2-NDC80 at various distances between the calponin homology (CH) domain of Hec1/Ndc80 protein (the NDC80 complex's microtubule binding domain adjacent to mTurquoise2 [*Alushin et al., 2010*]) and FlAsH-labeled microtubules and simulated the fluorescence decay curves, which revealed that NDC80 more than 8 nm away from the kMT do not contribute to the short-lifetime FRET state (*Figure 1—figure supplement 5A and B*). Thus, FRET only results when the CH domain of Hec1 is very close to the surface of kMTs, consistent with the short-lifetime species being NDC80 complexes whose Hec1 CH domains are bound to kMTs, an interpretation further supported by biological perturbation experiments described below.

Even though FRET results only from NDC80 bound to kMTs, the measured FRET fraction is not identical to the fraction of NDC80 bound to kMTs because not all tubulin heterodimers are labeled

with TC-FlAsH. Using large scale Monte Carlo simulations of mTurquoise2-NDC80 bound to FlAsH-labeled microtubules, we generated fluorescence decay curves for various NDC80 binding fractions, and estimated the resulting NDC80 FRET fractions from a fit to a two-exponential decay (see *Figure 1—figure supplement 5C* and Supplemental experiments in Materials and methods). We found that the NDC80 FRET fraction increases linearly with the NDC80 binding fraction with a slope of 0.42 ± 0.08, indicating that 42% of attached mTurquoise2-NDC80 contribute to the short-lifetime FRET state (*Figure 1—figure supplement 5C*). Thus, the measured FRET fraction of 13% in *Figure 1D* corresponds to 31% of NDC80 complexes being bound to kMTs.

## NDC80-kMT binding is regulated in a chromosome-autonomous fashion throughout prometaphase

Using the FLIM-FRET measurements of NDC80-kMT binding, we first investigated how NDC80-kMT binding evolves over the course of mitosis. We found that the average NDC80-kMT binding gradually increases as mitosis progresses, with NDC80 FRET fraction rising from 7% in early prometaphase to 14% in late metaphase, and reaching about 18% in anaphase (corresponding to NDC80 binding fractions of 17% in prometaphase; 33% in late metaphase; and 43% in anaphase) (*Figure 2A*). This temporal change in NDC80-kMT binding may underlie the previously observed decrease in kMT turnover throughout mitosis (*Kabeche and Compton, 2013*; *Zhai et al., 1995*).

The change in the average NDC80-kMT binding over the course of error correction in prometaphase could be due to a cell cycle-dependent coordinated regulation of NDC80 across kinetochores (coordinated regulation), an independent modulation of NDC80 on different chromosomes (chromosome-autonomous regulation), or a combination of both. To investigate the contribution of chromosome-autonomous regulation, we sought to determine if different populations of kinetochores in prometaphase exhibit different NDC80-kMT binding. We compared the extent of the NDC80-kMT binding of kinetochores centered at the metaphase plate to those located off-centered (*Figure 2B*), and found that the centered kinetochores exhibit 2.0 ± 0.4 times higher NDC80 binding than the off-centered kinetochores. We next investigated how the NDC80-kMT binding of centered and off-centered kinetochores change with time in prometaphase. As mitosis progresses and the chromosomes align to the metaphase plate, the number of kinetochores in the center region increases while the number of kinetochores in the off-center region decreases (*Figure 2C*). NDC80-kMT binding continuously increases over time for the kinetochores located in the center region, but remains constant with the FRET fraction of ~7% for the kinetochores in the off-center region (*Figure 2D*). The observation of differences in NDC80 binding between different subpopulations of kinetochores strongly argues for the existence of chromosome-autonomous regulation, which might be modulated by tension, Aurora kinases A and B, pathways that control the conversion of lateral to end-on kMT attachments, or other factors (*Godek et al., 2015*; *DeLuca et al., 2018*). We speculate that the temporal increase in NDC80-kMT binding of centered kinetochores is due to the gradual decrease in the number of kinetochores with erroneous attachment that transiently lie on the metaphase plate (*Magidson et al., 2011*). The constant NDC80-kMT binding of off-centered kinetochores argues for a lack of temporal regulation of this subpopulation.

## NDC80-kMT binding is positively correlated with centromere tension

After demonstrating that different population of kinetochores exhibits different NDC80-kMT binding throughout prometaphase, we next investigated the factors contributing to chromosome-autonomous regulation of the interaction between NDC80 and kMTs. Aligned chromosomes in U2OS cells oscillate around the metaphase plate, with microtubules attached to the leading and trailing kinetochores primarily depolymerizing and polymerizing, respectively (*Tirnauer et al., 2002*; *Armond et al., 2015*) (*Figure 3A*). The distance between sister kinetochores (referred to as K-K distance) fluctuates during the oscillation (*Magidson et al., 2011*), as the centromere deforms in response to the dynamic change in tension (*Figure 3A*). Therefore, chromosome oscillation provides a window to study how NDC80 binding is related to kMT dynamics and centromere tension in a physiologically relevant condition.

We first asked whether NDC80 binding is different on leading and trailing kinetochores. We acquired time-lapse movies of 17 metaphase cells, tracked their kinetochores, identified sister kinetochores by their relative motions (*Figure 3B*), and quantified the NDC80 binding fraction in groups

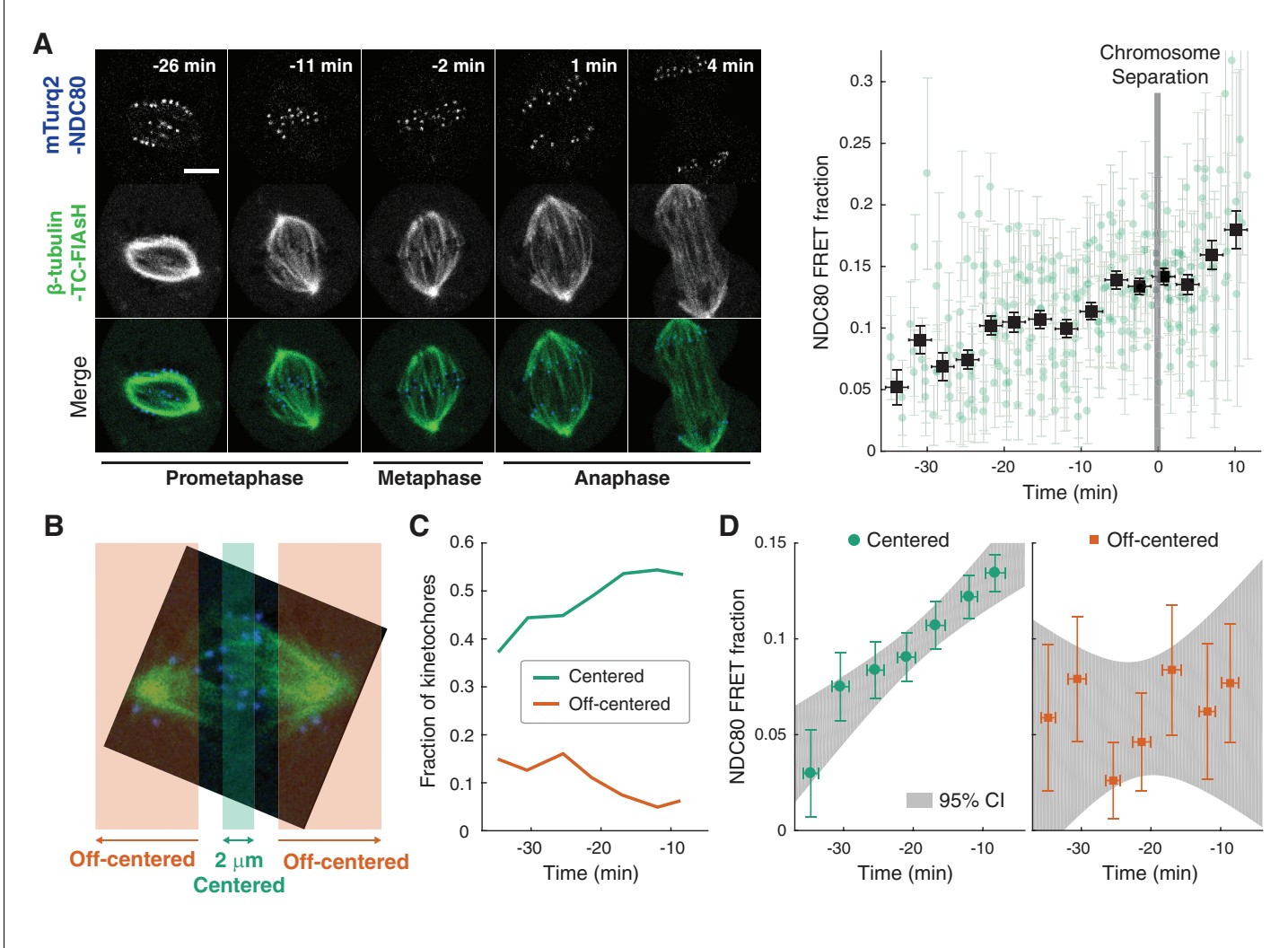

**Figure 2.** NDC80-kMT binding is regulated in a chromosome-autonomous fashion. (A) Example cell images and time course of NDC80 FRET fraction from prometaphase to metaphase to anaphase (n = 11 cells). Black squares are the mean, y-error bars are the SEM, and x-error bars are the SD of the data points (green circles) in equally spaced time intervals. 5 μm scale bar. (B) Kinetochores at each time point in prometaphase cells are divided into two groups, centered and off-centered, based on their distances from the metaphase plate. Kinetochores less than 1 μm away from the metaphase plate were classified as centered, and kinetochores more than 2.5 μm away were classified as off-centered. (C) Time course of the fraction of centered (green) and off-centered (orange) kinetochores in prometaphase. (D) Time course of NDC80 FRET fraction of centered (green circles) and off-centered (orange squares) kinetochores in prometaphase (n = 11 cells, 2886 centered and 572 off-centered kinetochores). Data points are the mean, y-error bars the SEM, and the x-error bars the SD in equally spaced time intervals. Gray areas are the 95% confidence intervals for the linear fits. Data points and source FLIM data are available in Figure 2-Data (*Yoo et al., 2018*).

DOI: https://doi.org/10.7554/eLife.36392.009

of kinetochores with similar velocities using FLIM-FRET analysis. We found that the NDC80 FRET fraction is higher at trailing kinetochores (12.8 ± 0.5%, SEM) than leading kinetochores (11.4 ± 0.5%, SEM), regardless of their speeds (*Figure 3C*), suggesting that NDC80 preferentially binds to polymerizing kMTs in vivo. The preferential binding is statistically significant (p<0.02, two-sided Z-test), yet small, presumably because leading and trailing kinetochores have a mixture of both polymerizing and depolymerizing MTs (*Armond et al., 2015*). This differential binding of NDC80 provides an explanation for the higher detachment rate of depolymerizing microtubules from kinetochores in vitro (*Akiyoshi et al., 2010*), and may give insight into the nature of kMT attachments (*Dumont et al., 2012*).

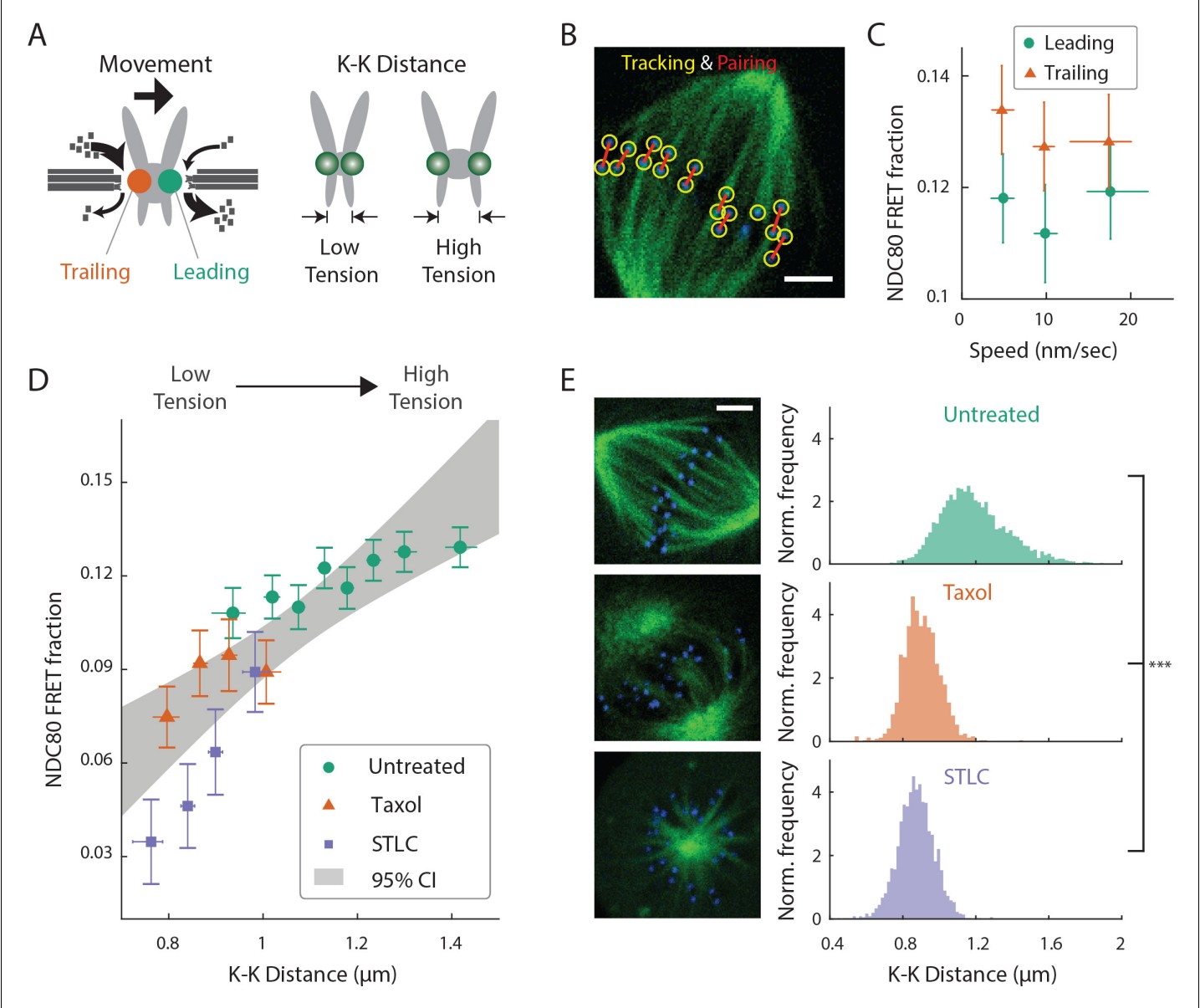

**Figure 3.** NDC80-kMT binding is correlated with kMT dynamics and centromere tension. (**A**) (left) kMTs predominantly depolymerize at leading kinetochores and polymerize at trailing kinetochores. (right) K-K distance is a proxy for centromere tension. Measuring NDC80-kMT binding along with the kinetochore movement and K-K distance therefore reveals how NDC80-kMT binding is related to the kMT dynamics and centromere tension. (**B**) Image of a metaphase cell with mTurquoise2-NDC80 (blue) and β-tubulin-TC-FlAsH (green), and kinetochore tracking (yellow circles) and pairing (red lines) results. 3 μm scale bar. (**C**) NDC80 FRET fraction vs. kinetochore speed for leading (green circle) and trailing (orange triangle) kinetochores (n = 17 cells, 681 kinetochores/data point). Data points are the mean, y-error bars the SEM, and the x-error bars the interquartile ranges within groups of kinetochores with similar velocities. (**D**) NDC80 FRET fraction vs. K-K distance for untreated cells (green circle, n = 17 cells, 984 kinetochores/data point), cells treated with 10 μM taxol (orange triangle, n = 7 cells, 525 kinetochores/data point), and cells treated with 5 μM STLC (purple square, n = 16 cells, 493 kinetochores/data point). For STLC data, only poleward-facing kinetochores are plotted (see *Figure 3—figure supplement 2* for comparison between poleward and anti-poleward kinetochores). Data points are the mean, y-error bars the SEM, and the x-error bars the interquartile ranges within groups of kinetochores with similar K-K distances. Gray area is the 95% confidence interval for the linear fit to the combined data. (**E**) Histograms of K-K distances for the untreated (top, green), taxol-treated (middle, orange), and STLC-treated (bottom, purple) cells. 3 μm scale bar in the cell images of mTurquoise2-NDC80 (blue) and beta-tublin-TC-FlAsH (green). ***p-value (Welch's t-test) less than $10^{-30}$. Data points and source FLIM data are available in Figure 3-Data (*Yoo et al., 2018*).

DOI: https://doi.org/10.7554/eLife.36392.010

The following figure supplements are available for figure 3:

**Figure supplement 1.** K-K distance and kinetochore velocity are not correlated.

*Figure 3 continued on next page*

*Figure 3 continued*

DOI: https://doi.org/10.7554/eLife.36392.011

**Figure supplement 2.** NDC80 FRET fraction of poleward- and anti-poleward-facing kinetochores in STLC-treated cells with monopolar spindles.

DOI: https://doi.org/10.7554/eLife.36392.012

The detachment rate of kMTs from kinetochores was shown to be reduced when tension was increased using glass needles in classic micromanipulation experiments by Bruce Nicklas (*Nicklas and Koch, 1969*). Since the NDC80 complex is the predominant coupler of the kinetochore to microtubules (*Cheeseman et al., 2006*; *DeLuca et al., 2006*), we hypothesized that the tension-dependent detachment of kMTs results from tension-dependent NDC80-kMT binding. To test this possibility, we next investigated the correlation between NDC80 FRET fraction and centromere tension, inferred by K-K distance, during chromosome oscillations. We used FLIM-FRET analysis to measure the NDC80 binding in groups of sister kinetochores with similar K-K distances, and observed a highly significant positive correlation ($p < 0.005$) between NDC80 FRET fraction and K-K distance (*Figure 3D*). We observed no significant correlation between K-K distance and kinetochore velocity ($p = 0.75$), arguing that NDC80 binding is independently regulated by these two factors (*Figure 3—figure supplement 1*). In the absence of microtubules, the rest length of K-K distance in human cell is $0.73 \pm 0.04$ μm (*Tauchman et al., 2015*), significantly shorter than the K-K distances during metaphase oscillations. Thus, in order to investigate a wider range of K-K distance, we treated cells with taxol, a microtubule-stabilizing drug, which greatly reduced K-K distances ($0.90 \pm 0.10$ μm, taxol vs. $1.19 \pm 0.19$ μm, untreated, SD, $p < 10^{-30}$) (*Figure 3E*) as well as NDC80 FRET fraction (*Figure 3D*). As an alternative way to reduce the tension, we inhibited Eg5 with 5 μM S-trityl-L-cysteine (STLC) (*Skoufias et al., 2006*). STLC-treated cells form monopolar spindles with reduced K-K distances ($0.87 \pm 0.10$ μm, SD, $p < 10^{-30}$) (*Figure 3E*). In these monopolar spindles, NDC80 FRET fractions from poleward-facing kinetochores was positively correlated with the K-K distance ($p < 0.05$), while anti-poleward-facing kinetochores displayed reduced NDC80 FRET fraction with no significant correlation with K-K distance ($p = 0.46$), presumably because many of these kinetochores are monotelically attached from the poleward side (*Figure 3—figure supplement 2*). The correlation between NDC80 FRET fraction and K-K distance was similar between taxol- and STLC-treated cells ($p = 0.15$, see *Figure 3D*), arguing that the relation between NDC80 FRET fraction and K-K distance is insensitive to the precise perturbation used to reduce tension. Combining the data of untreated, taxol-treated and STLC-treated (only poleward-facing kinetochores) cells, we found that the NDC80 FRET fraction continually increases with K-K distance over the full range of K-K distance (positive correlation, $p < 0.00005$) (*Figure 3D*). The extent of variation of NDC80-kMT binding with K-K distance is comparable to the extent of variation over the course of mitosis, from prometaphase to anaphase onset, as well as the extent of difference between centered and off-centered kinetochores in late prometaphase (compare *Figure 3D* with 2A and D).

## Aurora B kinase regulates NDC80-kMT binding in a graded fashion in vivo

Aurora B kinase is one of the best characterized components of the error correction process, and the N-terminal tail of the Hec1/Ndc80 protein in the NDC80 complex is a known substrate of Aurora B kinase that contains nine phosphorylation sites (*Tanaka et al., 2002*; *Biggins et al., 1999*; *Cheeseman et al., 2006*; *DeLuca et al., 2006*; *Ciferri et al., 2008*; *Hauf et al., 2003*). Previous biochemistry experiments demonstrated that the phosphorylation state of Hec1 modulates its binding to microtubules in vitro (*Cheeseman et al., 2006*; *Zaytsev et al., 2014*, *2015*). We used our FLIM-FRET technique to investigate the relationship between Aurora B kinase activity and NDC80-kMT binding in cells. We first added the ATP-competitive Aurora B inhibitor, ZM447439, to late prometaphase cells, and observed a gradual increase in NDC80 FRET fraction over ~10 min, from 9% to nearly 18% (corresponding to 21% NDC80 binding fraction before Aurora B inhibition and 43% after the inhibition) (*Figure 4A* and *Figure 4—figure supplement 1A*). Thus, Aurora B is a major modulator of NDC80-kMT binding in cells. This modulation of NDC80-kMT binding could occur directly, through phosphorylation of the N-terminal tail of the Hec1/Ndc80 protein, or indirectly, through other Aurora B substrates which are at kinetochores or which influence spindle assembly

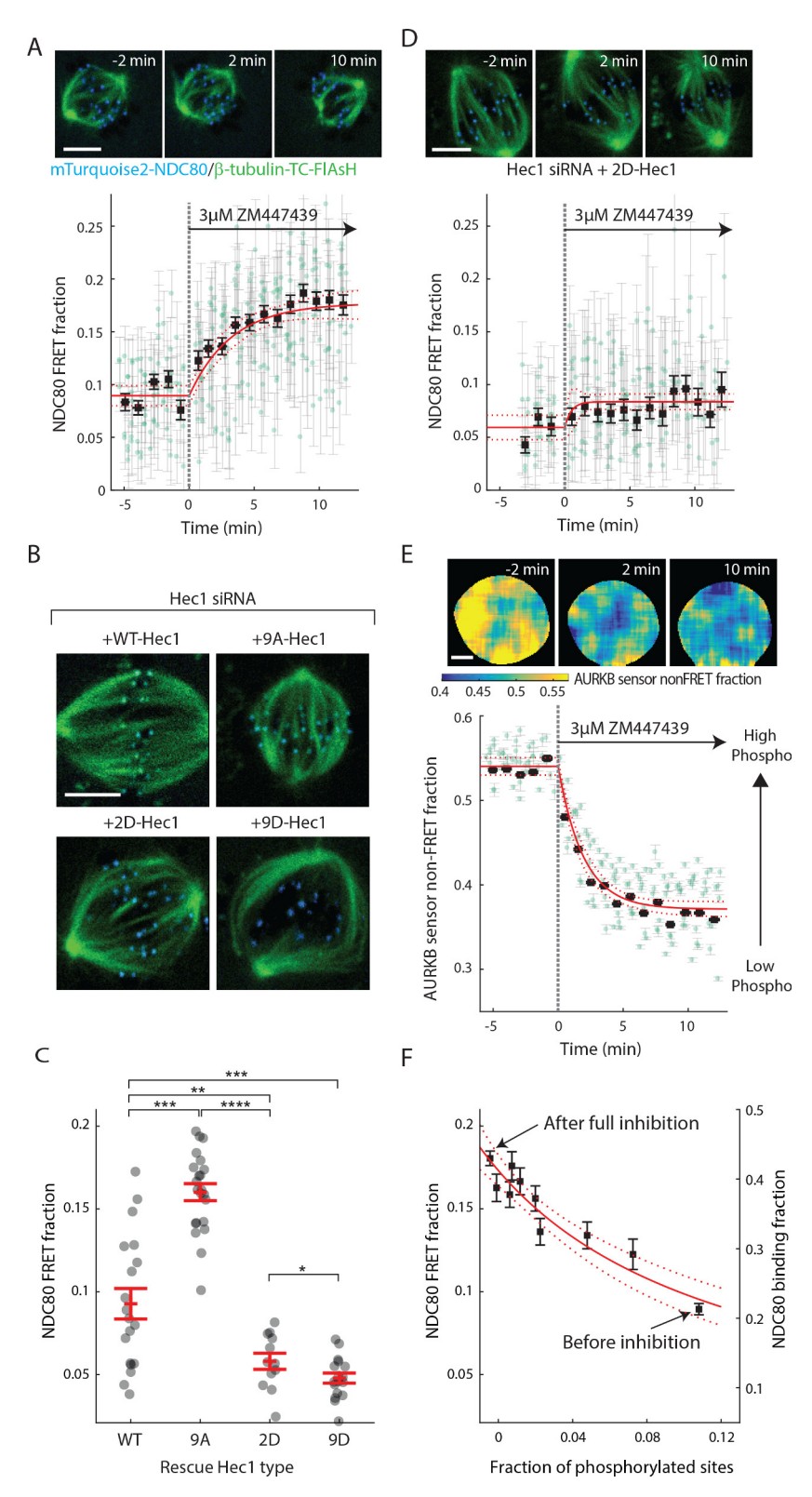

**Figure 4.** Aurora B kinase regulates NDC80-kMT binding in a graded fashion in vivo. (**A**) (top) Cell images showing mTurquoise2-NDC80 (blue) and beta-tubulin-TC-FlAsH (green). (bottom) Time course of NDC80 FRET fraction in response to Aurora B inhibition by 3 μM ZM447439 (n = 15 cells). (**B**) Images of cells with mTurquoise2-NDC80 (blue) and beta-tubulin-TC-FlAsH (green) after depleting endogenous Hec1 by siRNA and expressing siRNA-insensitive WT or three different phosphomimetic mutants of Hec1: 9A-, 2D-, and 9D-Hec1 (see Materials and methods). (**C**) NDC80 FRET fraction of
*Figure 4 continued on next page*

*Figure 4 continued*

cells whose endogenous Hec1 are replaced with WT or phosphomimetic Hec1 (see Materials and methods). Black dots are from individual cells and red error bars are mean ± SEM. n = 19, 22, 12, and 17 cells for WT-, 9A-, 2D-, and 9D-Hec1. *p<0.1; **p<0.01; ***p<0.001; ****p<0.0001. (D) Time course of NDC80 FRET fraction of 2D-Hec1-expressing cells in response to Aurora B inhibition by 3 µM ZM447439 (n = 12 cells). (E) (top) Cell images color-coded with Aurora B sensor non-FRET fraction. (bottom) Time course of the non-FRET fraction of the cytoplasmic Aurora B FRET sensor in response to 3 µM ZM447439 (n = 10 cells). In (A), (D) and (E), black squares and error bars are the weighted mean and SEM of the data points (green circles) in equally spaced time intervals of 1 min. Red solid and dashed lines are the best-fit exponential decay models and their 95% confidence intervals, respectively. 5 µm scale bar for all images. (F)NDC80 FRET fraction (from (A)) and NDC80 binding fraction (converted from the FRET fraction) plotted against the fraction of phosphorylated Aurora B phosphorylation sites in NDC80 (converted from Aurora B FRET sensor non-FRET fraction in (E)). Red solid and dashed lines are the best-fit NDC80-kMT binding model (derived in Mathematical modeling in Materials and methods) and its 95% confidence interval. Data points and source FLIM data are available in Figure 4-Data (*Yoo et al., 2018*).

DOI: https://doi.org/10.7554/eLife.36392.013

The following figure supplement is available for figure 4:

**Figure supplement 1.** Supplemental data for Aurora B inhibition experiments.

DOI: https://doi.org/10.7554/eLife.36392.014

(*Carmena et al., 2012*; *Krenn and Musacchio, 2015*). To further investigate the Aurora B modulation of NDC80 binding, we used non-phosphorylatable mutants of Hec1/Ndc80 protein, in which all nine identified Aurora B target sites are mutated to either aspartic acid (a phospho-mimicking mutation) or alanine (a phospho-null mutation) (*DeLuca et al., 2011*). We replaced the endogenous Hec1 with wild-type Hec1 or the phosphomimetic mutant of Hec1 by RNAi knockdown and rescue (*Figure 4B*, see Materials and methods). The NDC80 FRET fraction decreased with the number of phospho-mimicking mutations, from 16.0 ± 0.5% with 9A-Hec1 (all nine phosphorylation sites substituted with Ala) to 5.8 ± 0.5% with 2D-Hec1 (two sites, S44 and S55, substituted with Asp while the others with Ala) and to 4.8 ± 0.3% with 9D-Hec1 (all nine sites substituted with Asp) (mean ± SEM, *Figure 4C*). The average NDC80 FRET fraction of WT-Hec1 (9.3 ± 0.9%) was similar, but slightly higher than 2D-Hec1, consistent with previous results arguing that on average there are zero to two sites phosphorylated per Hec1/Ndc80 protein in prometaphase and metaphase (*Zaytsev et al., 2014*).

We next sought to determine if the increased binding of NDC80 to kMTs upon Aurora B inhibition is caused by a change of phosphorylation of the N-terminal tail of Hec1/Ndc80 protein. To this end, we added the Aurora B inhibitor, ZM447439, to cells with endogenous Hec1 replaced with 2D-Hec1, whose nine identified Aurora B target sites in the N-terminal tail cannot be phosphorylated. After the addition of ZM447439, the NDC80 FRET fraction increases by only 0.02 ± 0.01 (SE), which is statistically significant (p<0.001), but substantially smaller than the increase observed in cells with endogenous Hec1 (0.09 ± 0.01 (SE)) (compare *Figure 4D–4A*). Thus, the modulation of NDC80 binding to kMTs by Aurora B predominantly occurs through the phosphorylation of the N-terminal tail of Hec1/Ndc80.

We next quantified how Aurora B inhibition influences Aurora B activity in cells. We performed FLIM measurement on a cytoplasmic Aurora B FRET biosensor (*Fuller et al., 2008*), which contains a kinesin-13 family Aurora B substrate whose phosphorylation obstructs intramolecular FRET between mTurquoise2 and YPet (*Figure 4—figure supplement 1B*). During ZM447439 treatment, we found a continual reduction in the fraction of the Aurora B sensors in the non-FRET state, a proxy for Aurora B phosphorylation, from 0.540 ± 0.007 (SEM) to 0.368 ± 0.012 (SEM) (*Figure 4E* and *Figure 4—figure supplement 1C*). Nuf2-targeted Aurora B sensor responded to the ZM447439 treatment with similar kinetics, arguing that the time scale of response to Aurora B inhibition is insensitive to the spatial location of the substrate (*Figure 4—figure supplement 1D*). As the typical time scale of drug uptake is far slower than typical phosphorylation/dephosphorylation kinetics (*Thurber et al., 2014*; *Huang et al., 1997*), it is reasonable to assume that the phosphorylation level of Aurora B substrate is at steady state at each time point, so plotting the measured NDC80 binding fraction (converted from FRET fraction) vs. phosphorylated level (converted from Aurora B sensor non-FRET fraction) at each time point reveals their relationship. This analysis showed a graded dependence of NDC80-kMT binding on phosphorylation (*Figure 4F*), which is consistent with the impact of phosphomimetic Hec1 mutants on NDC80 binding (compare *Figure 4C and F*) and results from previous in vitro assays (*Zaytsev et al., 2014*, *2015*). The increased NDC80-kMT binding after Aurora B inhibition

may underlie the reduction in detachment of kMTs from kinetochores after Aurora B inhibition, observed in photoactivation experiments (*Cimini et al., 2006*).

## Haspin-dependent centromere-localized Aurora B is responsible for the tension dependency of NDC80-kMT binding

Aurora B regulates kinetochore-microtubule interactions, but its contribution to the tension-dependent stabilization of kinetochore-microtubule attachments is controversial (*Campbell and Desai, 2013*; *Salimian et al., 2011*; *Akiyoshi et al., 2010*; *Liu et al., 2009*; *Tanaka et al., 2002*; *Zaytsev et al., 2016*; *Godek et al., 2015*; *Lampson and Cheeseman, 2011*; *Haase et al., 2017*). We next sought to determine if the correlation between NDC80-kMT binding and centromere tension that we observed (*Figure 3D*) is caused by the phospho-regulation of NDC80 by Aurora B. After replacing the endogenous Hec1 protein with 9A-Hec1, which cannot be phosphorylated on the nine mutated target sites, we no longer observed a significant correlation between NDC80 FRET fraction and K-K distance (p=0.20) (*Figure 5A*). 9A-Hec1-expressing cells displayed significantly larger K-K distance than unperturbed cells (1.36 ± 0.21 μm, SD, p<10$^{-30}$) (*Figure 5B*), consistent with previous studies (*Tauchman et al., 2015*; *Zaytsev et al., 2014*). To investigate a wider range of K-K distance, we treated cells with taxol or STLC (*Figure 5-supplement figure 1A*), as described above, and found no correlation over the full range of K-K distance (p=0.29, *Figure 5A and B*). Since the strong correlation between NDC80 binding and K-K distance (*Figure 3D*) is eliminated in the non-phosphorylatable Hec1 mutant 9A-Hec1 (*Figure 5A*), this argues that the phosphorylation of the N-terminal tail of Hec1 is responsible for the correlation between NDC80-kMT binding and K-K distance. As Aurora B is believed to be the primary kinase that phosphorylates the N-terminal tail, this further suggests that the activity of Aurora B is responsible for the correlation between NDC80-kMT binding and K-K distance.

Aurora B is localized to centromeres in prometaphase and metaphase (*Carmena et al., 2012*). We next investigated if this localization is important for the correlation between NDC80-kMT binding and K-K distance. We used the haspin kinase inhibitor, 5-iodotubercidin (5-ITu), which has previously been shown to compromise the recruitment of Aurora B to inner-centromeres (*Wang et al., 2012*) (*Figure 5C*). After 10 min of exposure of cells to 5-ITu, INCENP, a member of the chromosome passenger complex (CPC), which also includes Aurora B (*Carmena et al., 2012*), was drastically reduced at centromeres (*Figure 5D*). Treating cells with 5-ITu for over 15 min did not significantly alter the average K-K distance (1.16 μm ± 0.18 μm, 5-ITu vs. 1.19 μm ± 0.19 μm, untreated, SD) or the overall average fraction of NDC80 bound to kMTs (FRET fraction, 11.79 ± 0.02% 5-ITu vs. 11.87 ± 0.02%, untreated, SEM), but eliminated the correlation between K-K distance and NDC80-kMT binding (p=0.41, *Figure 5E and F*). In order to investigate a wider range of K-K distance, we treated cells with both taxol and 5-ITu or with both STLC and 5-ITu (*Figure 5—supplement figure 1B*), and found no correlation over the full range of K-K distance (p=0.96, *Figure 5E and F*). Our observation that the mislocalization of Aurora B from centromeres does not affect the K-K distance but compromises the correlation between NDC80 binding and K-K distance argues that the correlation shown in *Figure 3D* is due to tension causing an increase in NDC80-kMT binding, rather than NDC80-kMT binding causing an increase in tension. Moreover, the tension dependency of NDC80-kMT binding in human tissue culture cells depends on Aurora B recruitment to centromeres by haspin kinase, further arguing in favor of models in which phosphorylation by Aurora B plays a central role in chromosome autonomous error correction.

We were surprised that the average NDC80-kMT binding does not significantly change after mislocalizing Aurora B with 5-ITu (compare *Figures 3D* and *5E*, and see *Figure 5G*). This suggests that Aurora B can still act on NDC80 even after the concentration of Aurora B at centromeres is greatly reduced. Consistent with this hypothesis, the Aurora B activity at kinetochores assessed by Nuf2-targeted Aurora B FRET sensor was not changed by 5-ITu treatment (*Figure 5H*). Furthermore, when cells treated with 5-ITu were exposed to the Aurora B inhibitor ZM447439, NDC80-kMT binding increased (to NDC80 FRET fraction of 0.17 ± 0.01, SEM) and Aurora B activity at kinetochores decreased (to Aurora B FRET sensor non-FRET fraction of 0.55 ± 0.01, SEM), indistinguishable from the levels in cells not subject to 5-ITu exposed to ZM447439 (*Figure 5G and H*). Thus, tension dependency of NDC80-kMT binding is conferred by Aurora B recruited to centromeres through a haspin-dependent pathway, while the average level of NDC80-kMT binding is also set by Aurora B, but in a manner that is not dependent on haspin.

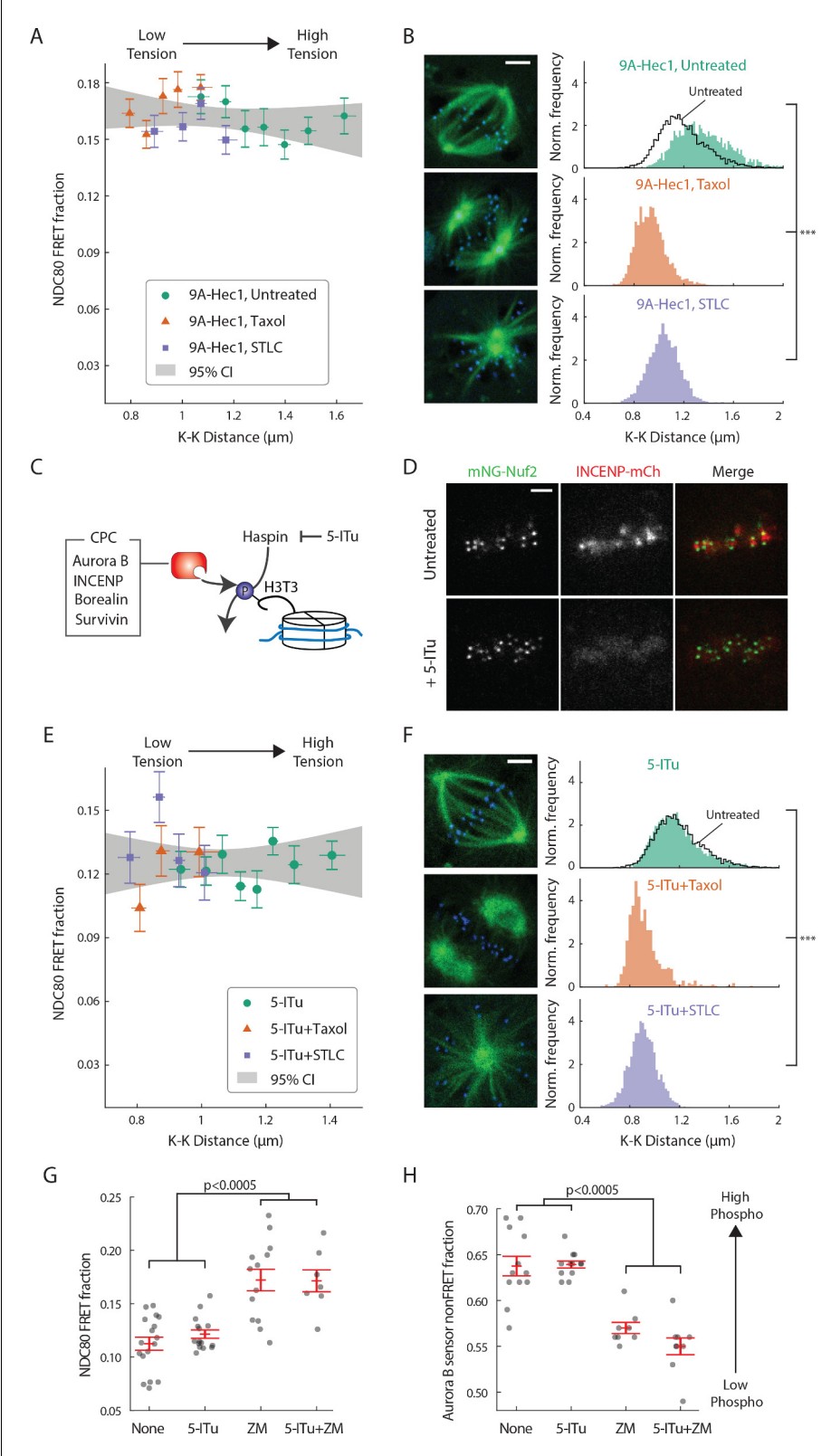

**Figure 5.** Haspin-dependent centromere-localized Aurora B is responsible for the tension dependency of NDC80-kMT binding. (**A**) NDC80 FRET fraction vs. K-K distance for 9A-Hec1-expressing cells with no drug treatment (green circle, n = 12 cells, 803 kinetochores/data point), with 10 μM taxol treatment (orange triangle, n = 9 cells, 1113 kinetochores/data point), or with 5 μM STLC treatment (purple square, n = 10 cells, 855 kinetochores/data point). For STLC data, only poleward-facing kinetochores are included (see *Figure 5—figure supplement 1A* for comparison between poleward-facing

*Figure 5 continued on next page*

*Figure 5 continued*

and anti-poleward-facing kinetochores). Data points are the mean, y-error bars the SEM, and the x-error bars the interquartile ranges within groups of kinetochores with similar K-K distances. Gray area is the 95% confidence interval for the linear fit to the combined data. (B) Histograms of K-K distances for the untreated 9A-Hec1 cells (top, green), untreated cells with endogenous Hec1 (top, black line), 9A-Hec1 cells treated with taxol (middle, orange), and 9A-Hec1 cells treated with STLC (bottom, purple). 3 µm scale bar in the cell images of mTurquoise2-NDC80 (blue) and beta-tubulin-TC-FlAsH (green). ***$p<10^{-6}$ (Welch's t-test). (C) Haspin kinase phosphorylates histone H3 at Thr3 (H3T3), which recruits the chromosome passenger complex (CPC, red) to centromeres. 5-Iodotubercidin (5-ITu) inhibits haspin kinase, thereby displacing Aurora B from centromeres. (D) Spinning-disk confocal microscopy images of cells expressing mNeonGreen-Nuf2 (green) and INCENP-mCherry (red) before (top) and after (bottom) haspin inhibition by 10 µM 5-ITu treatment. 3 µm scale bar. (E) NDC80 FRET fraction vs. K-K distance for cells treated with 10 µM 5-ITu (green circle, n = 15 cells, 1170 kinetochores/data point), for cells treated with both 10 µM 5-ITu and 10 µM taxol (orange triangle, n = 3 cells, 359 kinetochores/data point), and for cells treated with 10 µM 5-ITu and 5 µM STLC (purple square, n = 12 cells, 564 kinetochores/data point). For 5-ITu + STLC data, only poleward-facing kinetochores are included (see *Figure 5—figure supplement 1B* for comparison between poleward-facing and anti-poleward-facing kinetochores). Data points are the mean, y-error bars the SEM, and the x-error bars the interquartile ranges within groups of kinetochores with similar K-K distances. Gray area is the 95% confidence interval for the linear fit to the combined data. (F) Histograms of K-K distances for the 5-ITu-treated (top, green), untreated (top, black line), 5-ITu + taxol treated (middle, orange), and 5-ITu + STLC treated cells (bottom, purple). 3 µm scale bar in the cell images of mTurquoise2-NDC80 (blue) and beta-tubulin-TC-FlAsH (green). ***$p<10^{-6}$ (Welch's t-test). (G) NDC80 FRET fraction and (H) the non-FRET fraction of Nuf2-targeted Aurora B FRET sensor (proxy for Aurora B activity at NDC80) for different drug treatments. Each data point (gray circle) corresponds to an individual cell, and the error bar (red) shows the mean and SEM. P-values from two-sided Welch's t-test. Data points and source FLIM data are available in Figure 5-Data (*Yoo et al., 2018*).

DOI: https://doi.org/10.7554/eLife.36392.015

The following figure supplement is available for figure 5:

**Figure supplement 1.** NDC80 FRET fraction of poleward and anti-poleward kinetochores in STLC-induced monopolar spindles of 9A-Hec1-expressing cells and haspin inhibited cells.

DOI: https://doi.org/10.7554/eLife.36392.016

## The concentration of Aurora B at the location of NDC80 is dependent on centromere tension

The extent to which Aurora B phosphorylates NDC80 depends on the activity of Aurora B and the concentration of Aurora B at NDC80. To further investigate how the haspin-dependent pool of Aurora B confers tension dependency to NDC80-kMT binding, we next examined how Aurora B localization depends on K-K distance. We used spinning-disk confocal microscopy to image mNeon-Green-Nuf2, to locate NDC80, and INCENP-mCherry, to measure the distribution of Aurora B. We localized NDC80 to sub-pixel accuracy and identified sister kinetochore pairs (see *Figure 6A* and Materials and methods). For each pair of kinetochores, we measured the intensity of INCENP-mCherry at the location of NDC80, normalized on a cell-by-cell basis. Plotting the intensity of INCENP-mCherry at NDC80 as a function of K-K distance revealed a highly significant anti-correlation ($p<10^{-4}$, *Figure 6B*). To explore a larger range of K-K distances, we treated cells with 10 µM taxol. Combining the data of untreated and taxol-treated cells, we found that the intensity of INCENP-mCherry at NDC80 linearly decreases with K-K distance over the full range of K-K distance ($p<10^{-6}$). This observation suggests that the tension dependency of NDC80 binding may result from the decrease of Aurora B at NDC80 with increasing K-K distance.

We next investigated how the concentration of Aurora B at the location of NDC80 is influenced by haspin inhibition. In the presence of 10 µM 5-ITu, the concentration of Aurora B at NDC80 was greatly reduced and independent of K-K distance (*Figure 6B*). The lack of correlation between Aurora B concentration at NDC80 and K-K distance may explain the lack of tension dependency between NDC80-kMT binding and K-K distance upon haspin inhibition. We speculate that the finite concentration of Aurora B at NDC80 after haspin inhibition is the pool of Aurora B that maintains the average level of NDC80-kMT binding as described above.

## A biophysical model of tension dependent NDC80-kMT binding

Taking together, our data suggest that the concentration of Aurora B at NDC80 determines the extent of NDC80 phosphorylation, which in turn determines the level of NDC80-kMT binding. To further explore this possibility, we sought to determine the relationship between Aurora B concentration at NDC80 and NDC80-kMT binding. We plotted the NDC80-kMT binding (converted from the NDC80 FRET fraction in *Figures 3D* and *5E*) vs. the Aurora B concentration (converted from the normalized INCENP-mCherry intensity in *Figure 6D*) for each K-K distance, both with and without

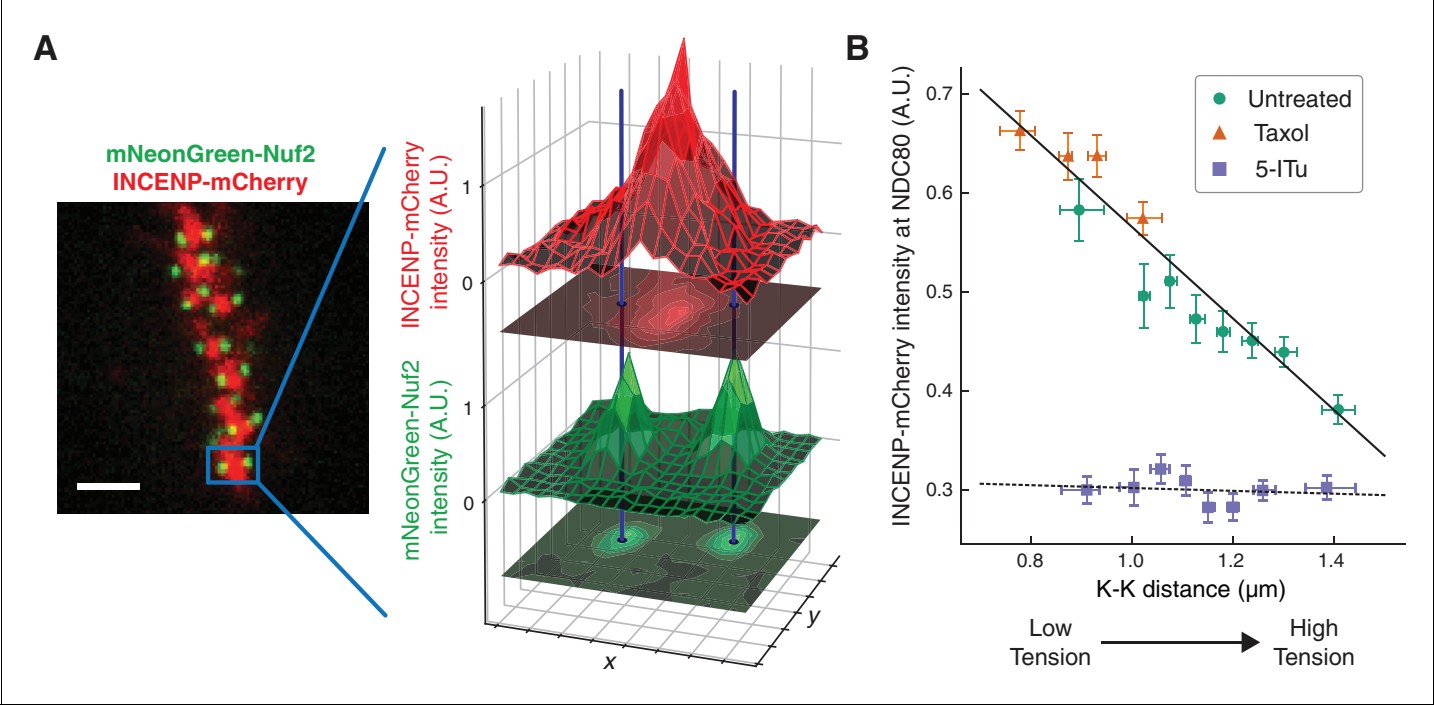

**Figure 6.** The concentration of Aurora B at the location of NDC80 decreases with centromere tension. (**A**) Spinning-disk confocal microscopy image of mNeonGreen-Nuf2 (green) and INCENP-mCherry (red). 3 µm scale bar. The location of NDC80 was determined to sub-pixel accuracy, using the mNeonGreen-Nuf2 image. For each pair of sister kinetochores, the intensity of INCENP-mCherry at the location of NDC80 was measured and normalized on a cell-by-cell basis. (**B**) Normalized INCENP-mCherry intensity at the location of NDC80 were averaged within groups of kinetochores with similar K-K distances, and plotted against the K-K distances for untreated (green circles), taxol-treated (orange triangles), and 5-ITu-treated (purple squares) cells. Data points are the mean, y-error bars the SEM, and the x-error bars the interquartile ranges. Black solid and dotted lines are the linear fits to DMSO+taxol combined data and 5-ITu data, respectively. 906 kinetochore pairs in 9 cells, 599 pairs in 8 cells, and 680 pairs in 6 cells were analyzed for DMSO control, taxol treatment, and 5-ITu treatment data, respectively. Data points and source FLIM data are available in Figure 6-Data (*Yoo et al., 2018*).

DOI: https://doi.org/10.7554/eLife.36392.017

haspin inhibition (*Figure 7A*, see Materials and methods). This revealed a highly nonlinear relationship: when the Aurora B concentration is lower than ~5 µM, the NDC80 binding fraction is independent of the Aurora B concentration, while for higher concentrations, the NDC80 binding fraction decreases with the Aurora B concentration (*Figure 7A*). We constructed a mathematical model to determine if this nonlinear relationship can be explained by the known biochemistry of Aurora B and NDC80 (*Figure 7A*) and observed change in the concentration of Aurora B at NDC80 (*Figure 6B*). In this model, we assume that there are two independent pools of Aurora B, haspin-dependent and haspin-independent, both of which engage in intermolecular autoactivation by phosphorylation in trans (*Zaytsev et al., 2016*; *Xu et al., 2010*; *Sessa et al., 2005*; *Bishop and Schumacher, 2002*), and are inactivated by phosphatases (*Zaytsev et al., 2016*; *Sessa et al., 2005*; *Kelly et al., 2007*; *Rosasco-Nitcher et al., 2008*). The activated Aurora B phosphorylates NDC80, which changes the binding affinity of NDC80 for kMTs (*Cheeseman et al., 2006*; *Zaytsev et al., 2014*; *Zaytsev et al., 2015*). This model can be solved analytically, and is sufficient to account for the relationship between NDC80 phosphorylation and NDC80-kMT binding (*Figure 4F*), and the relationship between Aurora B concentration at NDC80 and NDC80-kMT binding (*Figure 7A*). In this model, the nonlinear relationship between Aurora B concentration at NDC80 and NDC80-kMT binding ultimately results from the activation dynamics of Aurora B: at low concentrations, dephosphorylation by phosphatases overwhelm the in trans autoactivation, but above a threshold Aurora B concentration, A*, these two processes balance, leading to steady state level of activated Aurora B that further increases with increasing Aurora B concentration.

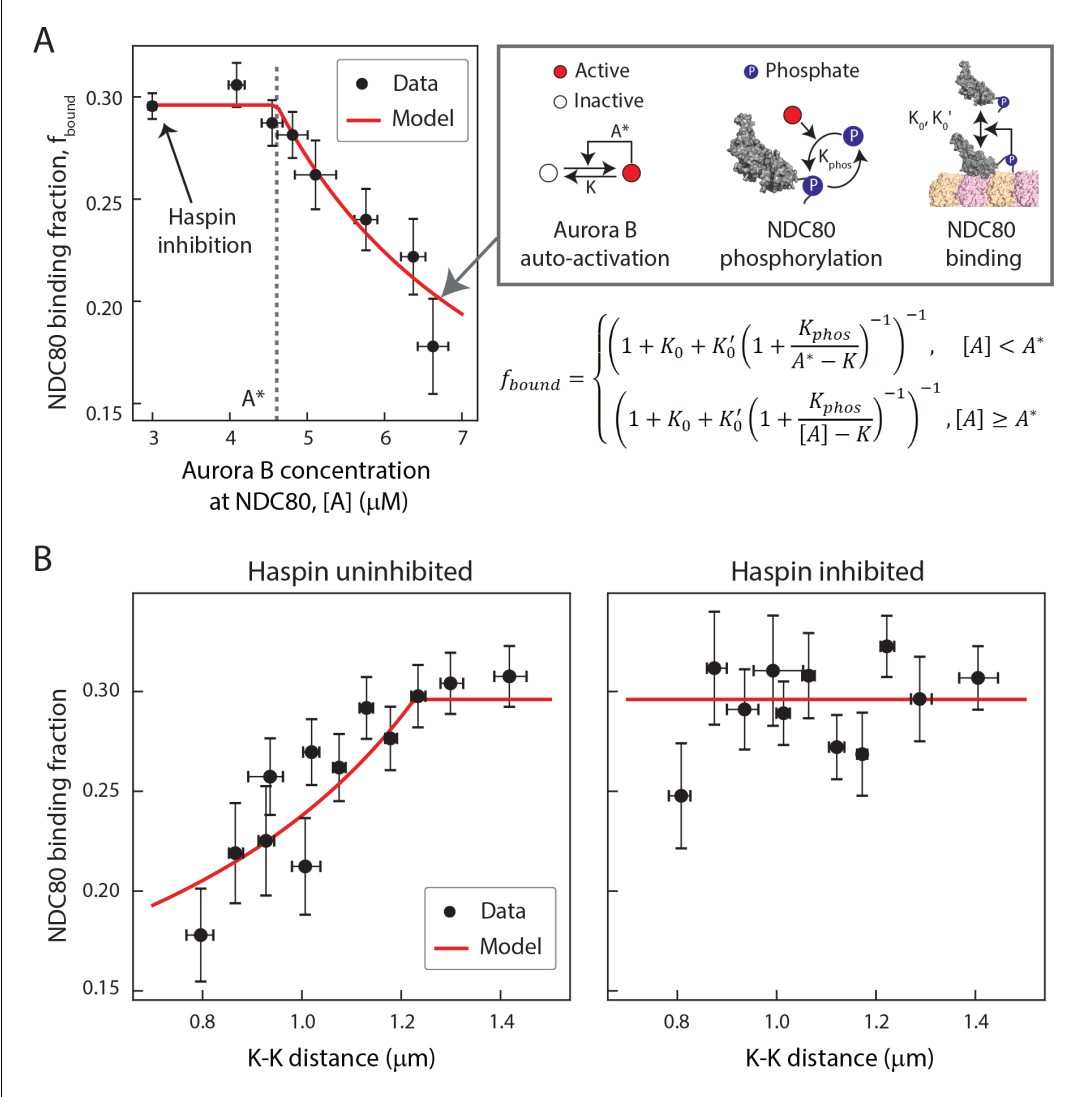

**Figure 7.** A biophysical model of tension dependent NDC80-kMT binding. (**A**) Plot of NDC80 binding fraction, $f_{bound}$, (converted from NDC80 FRET fraction in **Figures 3D** and **5E**) vs. Aurora B concentration at NDC80, [A] (converted from INCENP-mCherry intensity in **Figure 6B**, see Materials and methods). Data points (black circles) are the mean and SEM. We constructed a mathematical model that predicts NDC80 binding fraction from Aurora B concentration at NDC80 through three steps: intermolecular Aurora B auto-activation, NDC80 phosphorylation, and NDC80-kMT binding. Red line shows the mathematical model fit to the data. (**B**) NDC80 binding fraction vs. K-K distance before (left) and after (right) haspin inhibition by 5-ITu. The data points (black circles) are adapted from **Figures 3D** and **5E**. Red lines are the predictions from the mathematical model. Data points are available in Figure 7-Data (**Yoo et al., 2018**).

DOI: https://doi.org/10.7554/eLife.36392.018

We next investigated if the same model can recapitulate the tension dependency of NDC80-kMT binding. Inputting the measured linear relationship between Aurora B concentration at NDC80 and K-K distance (**Figure 6B**) into the model reproduced the observed tension-dependent behavior of NDC80-kMT binding (**Figure 7B**, left). Performing a similar procedure with the data from haspin inhibited cells (**Figures 5E** and **6B**) revealed that the model successfully predicts both the level of NDC80-kMT binding upon haspin inhibition and its independence on K-K distance (**Figure 7B**, right). Thus, this model provides a self-consistent, quantitative explanation of how the tension dependency of NDC80-kMT binding results from the biochemistry of Aurora B and NDC80, and the measured change in Aurora B concentration at NDC80 with tension, without the need to invoke diffusible gradients or additional mechanochemistry.

## Discussion

In this study, we developed a method to quantitatively measure the binding of the NDC80 complex to microtubules at individual kinetochores in human tissue culture cells. Our method uses TCSPC FLIM-FRET which, in contrast to intensity-based FRET, allows quantitative measurements of the fraction of molecules engaging in FRET, even with spatially varying concentrations of donors and acceptors. We calibrated our measurements using control experiments and Monte Carlo simulations, allowing us to convert the fraction of donor-labeled Nuf2 engaged in FRET to the fraction of NDC80 complexes bound to kMTs. This technique can be extended to the quantitative assessment of other protein-protein interactions in living cells.

Using this technique, we demonstrated that NDC80-kMT binding is regulated during prometaphase in a chromosome-autonomous manner. We observed that NDC80-kMT binding is strongly correlated to centromere tension, to an extent which is sufficient to account for the changes in NDC80-kMT binding over the course of prometaphase and metaphase. We characterized how Aurora B modulates NDC80-kMT binding in cells, which we found predominantly occurs through the phosphorylation of the N-terminal tail of Hec1. We showed that the correlation between NDC80-kMT binding and centromere tension is dependent on the phosphorylation of the N-terminal tail of Hec1. We determined the concentration of Aurora B at the locations of NDC80, which decreases with increasing centromere tension. Mislocalizing Aurora B by inhibiting haspin kinase eliminated the tension dependency of NDC80-kMT binding, but did not change its average level. The observation that inhibiting haspin removes the correlations between NDC80-kMT binding and tension, and between Aurora B localization and tension, but does not affect the distribution of K-K distances, argues that these correlations are caused by the influence of tension on NDC80-kMT binding and Aurora B localization. A simple mathematical model of Aurora B autoactivation and NDC80 phosphorylation and binding can quantitatively explain these results. Taking together, this leads to a biophysical model of the tension dependency of NDC80-kMT interactions, which arises from the nonlinearity of Aurora B autoactivation and the change in Aurora B concentration at NDC80 with centromere tension.

The FLIM-FRET technique developed in this study measures the fraction of NDC80 complexes whose Hec1 CH domains are bound to kMTs. Hypothetically, changes in NDC80 binding fraction might result from either alterations in the binding affinity of NDC80 or variations in the number of kMTs. Our results argue that the increase in NDC80 binding with increasing tension is caused by changes in affinity, because: (1) the addition of taxol causes a reduction in tension and a reduction in NDC80 binding (*Figure 3D and E*), but an increase in the number of kMTs (*McEwen et al., 1997*); (2) taxol-treated and STLC-treated cells exhibit the same correlation between tension and NDC80 binding (*Figure 3D*), suggesting that the decrease in binding with decreasing tension occurs independently of the mechanism of perturbation. Furthermore, previous estimates have found that each kMT can be contacted by approximately 40 NDC80 complexes (*Zaytsev et al., 2014*), and since there are approximately 240 NDC80 complexes per kinetochore (*Suzuki et al., 2015*), roughly 6 kMTs are sufficient for every NDC80 complex to be within reach of potential binding sites. Thus, NDC80 binding is expected to be insensitive to the number of kMTs if there are at least 6 kMTs. As on average there are ~20 kMTs per kinetochore in human cells (*Maiato et al., 2004*; *Rieder, 1982*), small changes in the number of kMTs are unlikely to modify the fraction of NDC80 complexes bound to kMTs. However, in circumstances where there are very few kMTs, for example in early prometaphase, changes in the number of kMTs will lead to changes in NDC80 binding. We hypothesize that the observed increase in NDC80 binding over the course of prometaphase to metaphase results from the combination of changes in kMT number and the changes in NDC80 binding affinity, due to the continual increase in tension at those times (*Magidson et al., 2011*). The difference in NDC80 binding between poleward-facing and anti-poleward-facing kinetochores in STLC-treated cells is likely caused by the difference in kMT numbers (*Figure 3—figure supplement 2* and *Figure 5—figure supplement 1*). This effect is particularly likely to explain the difference in NDC80 binding between sister kinetochores with 9A-Hec1, since these sister kinetochores are expected to exhibit the same NDC80 binding affinity (*Figure 5—figure supplement 1A*).

Error correction is believed to result from the regulation of the detachment of kMTs from kinetochores (*Godek et al., 2015*). As NDC80 is the primary coupler of kinetochores to microtubules (*Cheeseman et al., 2006*; *DeLuca et al., 2006*), it is reasonable to hypothesize that the rate of kMT

detachment from kinetochores might largely be governed by NDC80-kMT binding. Consistent with this, previous work showed that mutating NDC80 changes the number of kMTs in a manner that argues that increasing NDC80-kMT binding increases the stability of kMTs (*Guimaraes et al., 2008*; *Zaytsev et al., 2014*). Our work further supports the connection between NDC80-kMT binding and kMT stability by comparing our results with previous measurements of the rate of kMT detachment from kinetochores: NDC80-kMT binding increases during mitotic progression (*Figure 2*), while kMT stability increases (*Zhai et al., 1995*; *Kabeche and Compton, 2013*); NDC80-kMT binding increases in response to Aurora B inhibition (*Figure 4*), which causes an increase in kMT stability (*Cimini et al., 2006*); NDC80 preferentially binds to polymerizing kMTs over depolymerizing kMTs (*Figure 3C*), while reconstituted kinetochores bind more strongly to polymerizing microtubules than depolymerizing microtubules (*Akiyoshi et al., 2010*); NDC80-kMT binding increases with increasing tension (*Figure 3D*), and the stability of kMTs increase with increasing tension (*Nicklas and Koch, 1969*; *Akiyoshi et al., 2010*). These comparisons argue that the NDC80-kMT binding is a major determinant of the kMT detachment rate. Hence, we propose that tension dependency of kMT detachment from kinetochores, which is believed to underlie error correction, results from the tension dependency of NDC80-kMT binding. If correct, this implies that error correction ultimately results from the nonlinear autoactivation of Aurora B and the consequent phosphoregulation of NDC80-kMT binding. Further testing this proposal will require additional quantitative measurements of kMT detachment, errors, and error correction, in combination with measurement of NDC80-kMT binding using the FLIM-FRET method presented here.

# Materials and methods

## Key resources table

| Reagent type (species) or resource | Designation | Source or reference | Identifiers | Additional information |
|---|---|---|---|---|
| Cell line (*Homo sapiens*) | U2OS | ATCC | HTB-96 | |
| Transfected construct (*Homo sapiens*) | pBABE-puro mTurquoise2-Nuf2 | this paper | | Nuf2 N-terminally labeled with mTurquoise2; in retroviral vector with puromycin selection marker |
| Transfected construct (*Homo sapiens*) | pBABE-hygro mTurquoise2-Nuf2 | this paper | | Same as above, but with hygromycin selection marker |
| Transfected construct (*Homo sapiens*) | pBABE-blast mTurquoise2-Nuf2 | this paper | | Same as above, but with blasticidin marker |
| Transfected construct (*Homo sapiens*) | pBABE-blast Aurora B FRET sensor (mTurquoise2/YPet) | this paper | | modified from Addgene #45215; *Fuller et al. (2008)* |
| Transfected construct (*Homo sapiens*) | Nuf2-targeted Aurora B FRET sensor (mTurquoise2/Ypet) | this paper | | modified from Addgene #45215; *Fuller et al. (2008)* |
| Transfected construct (*Homo sapiens*) | mTurquoise2-TC | this paper | | mTurquoise2 with tetracysteine motif at the C-terminus |
| Transfected construct (Homo sapiens) | WT-Hec1-LSSmOrange | this paper | | modified from WT-Hec1-GFP from Jennifer DeLuca |
| Transfected construct (Homo sapiens) | 9A-Hec1-LSSmOrange | this paper | | modified from 9A-Hec1-GFP from Jennifer DeLuca |
| Transfected construct (Homo sapiens) | 2D(S44,55D)-Hec1 -LSSmOrange | this paper | | modified from 2D-Hec1-GFP from Jennifer DeLuca |
| Transfected construct (Homo sapiens) | 9D-Hec1-LSSmOrange | this paper | | modified from 9D-Hec1-GFP from Jennifer DeLuca |
| Transfected construct (*Homo sapiens*) | INCENP-mCherry | other | | Gift from Michael Lampson |
| Recombinant DNA reagent | pSpCas9(BB)—2A-GFP (pX458) | *Ran et al. (2013)* | Addgene: #48138 | |

*Continued on next page*

*Continued*

| Reagent type (species) or resource | Designation | Source or reference | Identifiers | Additional information |
|---|---|---|---|---|
| Sequence-based reagent | Donor single-stranded DNA for TC tag insertion at the C-terminus of TUBB | IDT | | ssDNA: cgtctctgagtatcagcagtacca ggatgccaccgcagaagaggaggaggattt cggtgaggaggccgaagaggaggcctGCT GTCCCGGCTGTTGctaaggcagagcccc catcacctcaggcttctcagttcccttagccgtc ttactcaactgcccctttcctctccctcaga; sgRNA target sequence: GAGGCCGAA GAGGAGGCCTA |
| Sequence-based reagent | Hec1 siRNA | Qiagen | Cat#: SI02653567 | |
| Peptide, recombinant protein | TC-peptide | Genscript | Custom designed | Synthesized, Ac-AEEEACCPGCC-NH2 |
| Commercial assay or kit | Amaxa Cell Line Nucleofector Kit V | Lonza | Cat#:VCA-1003 | |
| Commercial assay or kit | Ingenio Electroporation Kit | Mirus | Cat#: MIR 50118 | |
| Commercial assay or kit | Lipofectamine RNAiMax | Thermo Fisher | Cat#:13778075 | |
| Chemical compound, drug | FlAsH-EDT2 | Thermo Fisher | Cat#:T34561 | |
| Chemical compound, drug | 1,2-Ethanedithiol (EDT) | Alfa Aesar | Cat#:540-63-6 | |
| Chemical compound, drug | ZM447439 | Enzo Life Sciences | Cat#:BML-EI373 | |
| Chemical compound, drug | Paclitaxel (Taxol) | Enzo Life Sciences | Cat#:BML-T104 | |
| Chemical compound, drug | 5-iodotubercidin (5-ITu) | Enzo Life Sciences | Cat#:BML-EI29 | |
| Chemical compound, drug | S-Trityl-L-cysteine | Sigma Aldrich | Cat#:164739–5G | |
| Chemical compound, drug | Alexa Fluor 488 | Thermo Fisher | Cat#:A20000 | |
| Chemical compound, drug | Sodium 2-mercaptoethanesulfonate | Sigma Aldrich | Cat#:M1511 | |
| Software, algorithm | Interactive kinetochore FLIM-FRET analysis GUI (MATLAB 2016) | This paper | | http://doi.org/10.5281/zenodo.1198705; copy archived at https://github.com/ elifesciences-publications/FLIM-Interactive-Data-Analysis |
| Software, algorithm | Aurora B concentration at NDC80 analysis (Python 3) | This paper | | http://doi.org/10.5281/zenodo.1198702; copy archived at https://github.com/ elifesciences-publications/Aurora ConcentrationAnalysis |
| Software, algorithm | CAMPARI (v2) | Pappu Lab | | http://campari.sourceforge.net /V2/index.html |
| Software, algorithm | Rosetta 3.8 | RosettaCommons | | RRID:SCR_015701 |
| Other | 25 mm #1.5 poly-D-lysine coated round coverglass | neuVitro | Cat#:GG-25–1.5-pdl | |
| Other | FluoroBrite DMEM | Thermo Fisher | Cat#:A1896701 | |
| Other | Microtubule structure | *Zhang et al. (2015)* | PDB 3JAS | |
| Other | Human NDC80 bonsai decorated tubulin dimer | *Alushin et al. (2010)* | PDB 3IZ0 | |
| Other | mTurquoise structure | Stetten et al. (unpublished) | PDB 4B5Y | |

## Cell lines

U2OS cell lines (ATCC, HTB-96) were maintained in Dulbecco's modified Eagle's medium (DMEM, Thermo Fisher) supplemented with 10% Fetal Bovine Serum (FBS, Thermo Fisher), and 50 IU ml$^{-1}$ penicillin and 50 µg ml$^{-1}$ streptomycin (Thermo Fisher) at 37°C in a humidified atmosphere with 5% $CO_2$. Cells were validated as mycoplasma free by PCR-based mycoplasma detection kit (Sigma Aldrich).

## Live-cell imaging

All live-cell FLIM and spinning-disk confocal microscopy imaging were performed as follows. Cells were grown on a 25 mm diameter, #1.5-thickness, round coverglass coated with poly-D-lysine (GG-25–1.5-pdl, neuVitro) to 80~90% confluency. The cells were incubated in imaging media, which is FluoroBrite DMEM (Thermo Fisher) supplemented with 4 mM L-glutamine (Thermo Fisher) and 10 mM HEPES, for 15 ~ 30 min before imaging. The coverglass was mounted on a custom-built temperature controlled microscope chamber at 37°C, while covered with 1 ml of imaging media and 2 ml of white mineral oil (VWR). An objective heater (Bioptech) was used to maintain the objective at 37°C. We confirmed that the cells can normally divide longer than 6 hr in this condition. Only cells displaying proper chromosome alignment, normal spindle morphology, and high signal-to-noise ratio were selected for imaging and analysis.

## NDC80-kMT FLIM-FRET measurement mTurquoise2-NDC80/β-tubulin-TC-FlAsH stable U2OS cell line

A tetracysteine (TC) tag, CCPGCC, was genetically attached to the C-terminal end of tubulin beta class I (TUBB), an isotype of β-tubulin that is predominantly expressed in U2OS (assessed by qPCR, data not shown) and most other cancer cells (*Leandro-García et al., 2010*). The attachment of the TC tag was achieved by CRISPR-induced homologous recombination to ensure the consistent expression of labeled β-tubulin. ssDNA (IDT) with TC tag (5'-TGCTGTCCCGGCTGTTGC-3') and ~80 bp-long homology arms was used as a donor DNA. pSpCas9(BB)−2A-GFP (Addgene plasmid # 48138) (*Ran et al., 2013*) was utilized as a backbone for the plasmid carrying a sgRNA (5'-GAGGCC-GAAGAGGAGGCCUA-3') and Cas9. The plasmid and the donor ssDNA were simultaneously delivered into U2OS cells by electroporation (Nucleofector 2b and Amaxa Cell Line Nucleofector Kit V, Lonza). The insertion of the TC tag was verified through a PCR-based genotyping with primers 5'-GCATGGACGAGATGGAGTTCAC-3' and 5'-CCAGCCGTGTTTCCCTAAATAAG-3', qPCR, and a fluorescence imaging after FlAsH-EDT$_2$ staining.

The U2OS cells expressing TC-tagged β-tubulin were further engineered to stably express Nuf2 N-terminally labeled with mTurquoise2 (*Goedhart et al., 2012*) by retroviral transfection, three times with different antibiotic selections, 1 µg ml$^{-1}$ puromycin, 2 µg ml$^{-1}$ blasticidin, and 200 µg ml$^{-1}$ hygromycin (all from Thermo Fisher). The retroviral vectors and their information are available on Addgene (plasmid #: 80760, 80761, 80762). Monoclonal cell line was obtained by single cell sorting.

## FlAsH-EDT$_2$ staining

The protocol for the association of FlAsH-EDT$_2$ with β-tubulin-TC in cell was adapted from the previous study (*Hoffmann et al., 2010*) so as to maximize the labeling fraction while maintaining cell viability. The engineered U2OS cells expressing β-tubulin-TC were grown to 80~90% confluency in a 30 mm cell culture dish, and then were gently washed with Opti-MEM (Thermo Fisher) twice, and then stained in 2 ml Opti-MEM with 1 µM FlAsH-EDT$_2$ (Thermo Fisher) for 2 hr. To reduce the non-specific binding of FlAsH, the stained cells were subsequently incubated in Opti-MEM containing 250 µM 1,2-Ethanedithiol (EDT, Alfa Aesar) for 10 min, followed by a gentle wash with Opti-MEM. The cells were incubated in DMEM with 10% FBS for 6~10 hr before imaging, because they were found to be interphase-arrested for the first ~5 hr after the incubation with 250 µM EDT. Every buffers and media above were pre-warmed at 37°C before use. All incubation steps were performed at 37°C in a humidified atmosphere with 5% $CO_2$.

## FLIM measurement

Schematic instrumental setup of FLIM is shown in *Figure 1—figure supplement 2A*, and more details can be found in previous work (*Yoo and Needleman, 2016*). FLIM measurements were

performed on a Nikon Eclipse Ti microscope using two-photon excitation from a Ti:Sapphire pulsed laser (Mai-Tai, Spectral-Physics) with an 80-MHz repetition rate and ~70 fs pulse width, a galvanometer scanner (DCS-120, Becker and Hickl), TCSPC module (SPC-150, Becker and Hickl) and two hybrid detectors (HPM-100–40, Becker and Hickl). Objective piezo stage (P-725, Physik Instrumente) and motorized stage (ProScan II, Prior Scientific) were used to perform multi-dimensional acquisition, and a motor-driven shutter (Sutter Instrument) was used to block the excitation laser between acquisitions. The wavelength of the excitation laser was set to 865 nm. 470/24 and 525/30 bandpass emission filters (Chroma) were mounted on each detector, and a dichroic beam splitter (FF506-Di03, Semrock) was used for the simultaneous detection of mTurquoise2 and FlAsH fluorescence. The excitation laser was expanded to overfill the back-aperture of a water-immersion objective (CFI Apo 40 × WI, NA 1.25, Nikon). The power of the excitation laser was adjusted to 1.1~1.5 mW at the objective. All the electronics were controlled by SPCM software (Becker and Hickl) and µManager (*Edelstein et al., 2014*). Scanning area was set to either 13.75 × 13.75 µm or 27.5 × 27.5 µm, and the pixel size was set to 107 nm. Each image was acquired for 3~5 s of integration time. Acquisition interval was set to 13 s for the FLIM-FRET data in *Figures 3* and *5*, and 60~90 s for the FLIM-FRET data in *Figures 2* and *4*. Three or four z-sections, separated by 1 µm, were acquired for each time point. No photo-bleaching or photo-damage was observed in this imaging condition. Mitotic phases were judged by the arrangement of kinetochores.

## Kinetochore tracking and pairing

For the kinetochore FLIM-FRET measurements shown in *Figure 2–5*, custom-built MATLAB graphical user interphase (GUI) (available at http://doi.org/10.5281/zenodo.1198705 [*Yoo, 2018b*]; copy archived at https://github.com/elifesciences-publications/FLIM-Interactive-Data-Analysis) was used to import Becker and Hickl FLIM data, track kinetochores, identify kinetochore pairs, extract the FLIM curve from each kinetochore, and estimate the FLIM parameters using a nonlinear least-squared fitting or Bayesian FLIM analysis, as described below and in previous work (*Yoo and Needleman, 2016*). The GUI also allows the users to scrutinize and manually correct the kinetochore trajectories and pairing. The kinetochore tracking algorithm was adapted from a particle tracking algorithm (*Pelletier et al., 2009*), and the pair identification was performed by selecting pairs of kinetochores with distances and velocity correlations in predefined ranges. Drift correction was done by measuring correlation between two consecutive spindle images. The velocity $v(t)$ of kinetochore (in *Figure 3*) was estimated from the position $x(t)$ using the five-point method:

$$v(t) \approx \frac{-x(t+2\Delta t) + 8x(t+\Delta t) - 8(t-\Delta t) + x(t-2\Delta t)}{12\Delta t}$$

Leading and trailing kinetochores (in *Figure 3*) were determined based on the velocities and the relative positions of paired sister kinetochores. The metaphase plate (in *Figure 2*) was determined by finding an equidistant plane between the two spindle poles (that were manually located based on spindle images). Poleward-facing and anti-poleward-facing kinetochores (in *Figures 3* and *5*) in STLC-treated cells were determined based on the relative positions of paired sister kinetochores and the position of the spindle pole, which was approximated by the average of the positions of kinetochores in each time point.

## Bayesian FLIM analysis

Fluorescence decay curves from individual kinetochores at each time point contain only a few hundreds of photons. In this low photon count regime, FLIM analysis with conventional least-squared nonlinear regressions results in significantly biased estimate for the parameters (*Kaye et al., 2017*; *Rowley et al., 2016*). Therefore, we used a Bayesian approach, which has been described and tested previously (*Yoo and Needleman, 2016*; *Kaye et al., 2017*), and is briefly explained below.

Let $\theta$ be the set of parameters of the FLIM-FRET model, and $y = \{y_i\}$ be the observed FLIM data, where $y_i$ is the number of photons detected in the $i$-th time bin of the FLIM curve. Then the posterior distribution of $\theta$ (assuming a uniform prior distribution) is

$$p(\theta|y) \propto \prod_{i=1}^{N} P(t_{ar} \in [(i-1)\Delta t, i\Delta t]|\theta)^{y_i}$$

where $t_{ar}$ is the photon arrival time, and $N$ is the number of time bins. Since the size of the time bin ($\Delta t$, ~50 ps) is much smaller than the time scale of fluorescence decay (~ns), the probability that the arrival time $t_{ar}$ falls in the $i$-th time bin can be approximated by a Riemann sum:

$$P(t_{ar} \in [(i-1)\Delta t, i\Delta t] | \theta) \cong \sum_{k=(i-1)K+1}^{k=iK} h_\theta\left(k\widetilde{\Delta t}\right)\widetilde{\Delta t}$$

where $h_\theta$ is the discretized FLIM model, $\widetilde{\Delta t}$ is the size of time bin with which instrument response function (IRF) is measured, and the ratio $K = \frac{\Delta t}{\widetilde{\Delta t}}$ is the ADC ratio, which is set to 16 for our data. $h_\theta\left(k\widetilde{\Delta t}\right)$ can be written as the convolution between the IRF and an exponential decay model, $g_\theta$:

$$\begin{aligned} h_\theta\left(k\widetilde{\Delta t}\right) &= (IRF * (Ag_\theta + (1-A)))\left(k\widetilde{\Delta t}\right) \\ &\cong \sum_l mIRF\left[l - b_{shift}\right]\left(Ag_\theta\left((k-l)\widetilde{\Delta t}\right) + (1-A)\right) \end{aligned}$$

where $mIRF$ is the IRF measured with the finest time bins of size $\widetilde{\Delta t}$, and $b_{shift}$ is an integer parameter that determines the approximate shift of measured IRF relative to the theoretical IRF. $(1-A)$ indicates the relative contribution of noise that is uniformly distributed over time. The exponential decay model $g_\theta(t_d)$ is set to $\exp\left(-\frac{t_d}{\tau}\right)$ for the single-exponential decay model or $(1-f_{FRET})e^{-\frac{t_d}{\tau_D}} + f_{FRET}e^{-\frac{t_d}{\tau_{FRET}}}$ for the two-exponential decay model, where $0 \leq f_{FRET} \leq 1$ is the FRET fraction. The posterior distribution was computed by Gibbs sampling if the number of free parameters is greater than 3, or by grid sampling otherwise (for example, when both long and short lifetimes are fixed).

## NDC80 FRET fraction measurement procedures

The instrument response function (IRF) was acquired by measuring second-harmonic generation from a urea crystal. Negative control FLIM measurements on the engineered cells (mTurquoise2-NDC80/β-tubulin-TC) not incubated with FlAsH were performed for every experiment and the fluorescence decay curves extracted from kinetochores were analyzed with a single-exponential FLIM-FRET model to determine the long non-FRET lifetime, which is usually 3.7 to 3.8 ns. The short FRET lifetime was estimated by performing a two-exponential Bayesian FLIM-FRET analysis on the aggregated FLIM data of kinetochores in each cell stained with FlAsH while fixing the non-FRET lifetime to the value pre-determined from the negative control. Then we performed a two-exponential Bayesian FLIM-FRET analysis, with both FRET and non-FRET lifetimes fixed to the predetermined values, on FLIM data from each kinetochore. Kinetochores were grouped by time (*Figures 2* and *4*), positions (*Figure 2B–D*), velocities (*Figure 3C*), and K-K distances (*Figures 3D*, *5A and E*). The posterior distributions in a group of kinetochores were multiplied and then marginalized to obtain the mean and SEM of the FRET fraction. We previously confirmed that this way of combining posterior distribution gives an unbiased estimate of the mean FRET fraction (*Kaye et al., 2017*). NDC80 binding fraction was calculated by dividing NDC80 FRET fraction by the conversion factor 0.42, which had been determined by the calibration shown in *Figure 1—figure supplement 5C*.

## Aurora B kinase activity measurement

An Aurora B FRET sensor was constructed by replacing CyPet in a previous construct (Addgene plasmid # 45215) (*Fuller et al., 2008*) with mTurquoise2. The FRET sensor contains a kinesin-13 family Aurora B substrate whose phosphorylation results in its binding to the forkhead-associated domain in the sensor, which constrains the sensor to be in an open conformation and obstructs intramolecular FRET between mTurquoise2 and YPet (*Figure 4—figure supplement 1B*). Hence, the non-FRET fraction of the Aurora B FRET sensor is proportional to the Aurora B activity. The cytoplasmic Aurora B FRET sensor was stably expressed in U2OS cells by retroviral transfection (plasmid available on Addgene, plasmid # 83286). The Nuf2-targeted Aurora B FRET sensor was transiently transfected by electroporation (Nucleofector 2b, Lonza; Ingenio Electroporation Kit, Mirus) a day before imaging. The non-FRET fraction of the Aurora B FRET sensor was measured by FLIM-FRET in the same way as NDC80 FRET measurements described above. The exponential decay models $y_{\text{binding}}(t) = A\left(1 - \exp\left(-\frac{t_{t \geq 0}}{\tau}\right)\right) + c$ and $y_{\text{Aurora}}(t) = A\exp\left(-\frac{t_{t \geq 0}}{\tau}\right) + c$ were fitted to the time courses of NDC80

FRET fraction and FRET sensor non-FRET fraction after ZM447439, respectively (**Figure 4A,D and E** and **Figure 4—figure supplement 1D**), where $I_{t \geq 0}$ is equal to 0 if $t$ is less than zero, and 1 otherwise. The estimated parameter values are given in **Table 1**:

The fraction of Aurora B phosphorylation sites in NDC80, $f_{phos}$ (x-axis of **Figure 4F**), was converted from the non-FRET fraction of Aurora B FRET sensor, $f_{sensor}$ (y-axis of **Figure 4E**), as follows. First, we assumed that $f_{sensor}$ increases linearly with $f_{phos}$. Our result (**Figure 4C**) and previous work (**Zaytsev et al., 2014**) suggest that Ndc80 has about one phospho-residue out of nine phosphorylation sites in late prometaphase, based on which we assumed that $f_{phos}^{WT} = 1/9$ before Aurora B inhibition, and $f_{phos}^{ZM} = 0$ after the full Aurora B inhibition. Since $f_{sensor}$ were measured to be $f_{sensor}^{WT} = 0.540 \pm 0.007$ (SEM) before Aurora B inhibition and $f_{sensor}^{ZM} = 0.368 \pm 0.012$ (SEM) after the full Aurora B inhibition (**Figure 4E**), we converted $f_{sensor}$ to $f_{phos}$ by:

$$f_{phos} = \frac{f_{sensor} - f_{sensor}^{ZM}}{f_{sensor}^{WT} - f_{sensor}^{ZM}} \left( f_{phos}^{WT} - f_{phos}^{ZM} \right) + f_{phos}^{ZM} = 0.646(f_{sensor} - 0.368)$$

The $f_{bound}$ vs $f_{phos}$ data in **Figure 4F** was fit using a NDC80 binding model:

$$f_{bound} = \left( 1 + K_0 + K_0' f_{phos} \right)^{-1}$$

which is derived in Mathematical modeling section below.

## Phosphomimetic Hec1 mutant experiments

We used three different non-phosphorylable mutant Hec1 (gift from Jennifer DeLuca) in which all nine identified Aurora B target sites in the N-terminal tail are mutated to either Asp (phospho-mimicking mutation) or Ala (phospho-null mutation) (**DeLuca et al., 2011**; **Zaytsev et al., 2015**, **2014**): 9A-Hec1 (all nine sites substituted with Ala), 2D-Hec1 (two sites, S44 and S55, substituted with Asp, while the other seven sites with Ala), and 9D-Hec1 (all nine sites substituted with Asp). WT-Hec1 and the mutant Hec1 are C-terminally labeled with LSSmOrange. LSSmOrange signal at kinetochores were assessed to ensure the expression of the substituting Hec1 in cells.

Cells were grown to 50% confluence on a 10 cm petri dish in DMEM supplemented with 10% FBS andpenicillin-streptomycin (P/S) as described above. To knock down endogenous Hec1/Ndc80 protein, we used a FlexiTube siRNA duplex targeted to the 5' UTR of the Hec1 gene (5'-TCCCTGGG TCGTGTCAGGAAA-3', QIAGEN Hs_KNTC2_7 SI02653567). We incubated 240 pmol of the siRNA in 1.2 mL Opti-MEM (ThermoFisher 51985091) for 5 min with periodic flicking. We simultaneously incubated 8 µL of Lipofectamine RNAiMax (ThermoFisher 13778030) in 1.2 mL Opti-MEM for 5 min. We then combined the siRNA and Lipofectamine solutions and incubated at room temperature for 30

**Table 1.**

| Figure | Parameter | Mean | 95% CI |
|--------|-----------|------|--------|
| 4A | $A$ | 0.088 | (0.069,0.106) |
| | $\tau$ (min) | 3.26 | (1.31,5.21) |
| | $c$ | 0.089 | (0.080,0.099) |
| 4D | $A$ | 0.024 | (0.011,0.038) |
| | $\tau$ (min) | 0.50 | (−0.70,1.71) |
| | $c$ | 0.059 | (0.048,0.071) |
| 4E | $A$ | 0.17 | (0.16,0.18) |
| | $\tau$ (min) | 1.95 | (1.46,2.45) |
| | $c$ | 0.37 | (0.36,0.38) |
| 4-S1D | $A$ | 0.076 | (0.061,0.090) |
| | $\tau$ (min) | 1.12 | (0.23,2.00) |
| | $c$ | 0.56 | (0.55,0.57) |

DOI: https://doi.org/10.7554/eLife.36392.019

min with periodic flicking. Prior to adding the siRNA-lipid complex, we washed the cells once with PBS and then replaced the media with 8 mL Opti-MEM supplemented with 10% FBS. We then added the entire 2.4 mL siRNA mixture to the cells dropwise and incubated the cells at 37°C for 30 hr. Following the incubation, we nucleofected 2 µg of plasmid encoding WT-, 9A-, 2D-, or 9D-Hec1 along with an additional 30 pmol of Hec1 siRNA into 1 million cells using a Lonza Nucleofector 2b. We spread these cells evenly over three 35 mm dishes containing 25 mm poly-D-lysine coated coverslips and 2 mL of Opti-MEM supplemented with 10% FBS and P/S. We incubated overnight at 37°C for 18 hr before staining with TC-FlAsH and FLIM-FRET imaging as described above.

## Aurora B concentration at NDC80 measurement

### mNeonGreen-Nuf2/INCENP-mCherry U2OS cell

mNeonGreen fluorescent protein (*Shaner et al., 2013*) was genetically attached to the N-terminal end of Nuf2 by CRISPR-induced homologous recombination with an sgRNA (5'-GAAAGACAAAG TTTCCATCTTGG-3') and mNeonGreen sequence (Allele Biotechnology) flanked by 2 kb homology arms as a donor template. Monoclonal cell line was obtained by fluorescence-activated cell sorting and screened by fluorescent microscopy imaging. The mNeonGreen-Nuf2 U2OS cell line was transiently transfected with INCENP-mCherry (gift from Michael Lampson) by electroporation (Nucleofector 2b and Amaxa Cell Line Nucleofector Kit V, Lonza) a day before imaging, using the manufacturer's protocol.

### Spinning-disk confocal microscopy imaging

Cells were imaged using a spinning-disk confocal microscope (Nikon Ti2000, Yokugawa CSU-X1) with 1.5x magnification lens and 1.2x tube lens, an EM-CCD camera (Hamamatsu), a 60x water-immersion objective (Nikon), an objective piezo stage (P-725, Physik Instrumente), and motorized x-y stage (ProScan II, Prior Scientific) controlled by µManager (*Edelstein et al., 2014*). A 488 nm laser and 514/30 filter were used to image mNeonGreen-Nuf2, and a 560 nm laser and 593/40 filter were used to image INCENP-mCherry. 11–15 z-slices, separated by 2 µm, were taken for each time point. Three time points, separated by a minute, were acquired before and after DMSO (for untreated data), 10 µM taxol, or 10 µM 5-ITu treatment.

### Aurora B concentration at NDC80 measurement

Image analysis was performed by a custom Python code, available at http://doi.org/10.5281/zenodo.1198702 (*Yoo, 2018a*; copy archived at https://github.com/elifesciences-publications/Aurora-ConcentrationAnalysis). Kinetochore identification was achieved by applying *trackpy* package (github.com/soft-matter/trackpy) to mNeonGreen-Nuf2 fluorescence images. The sub-pixel location of NDC80 was calculated by centroid estimation. Sister kinetochore pairs were determined based on the relative positions of kinetochores and the INCENP-mCherry intensity between kinetochores. For each identified kinetochore pair, INCENP-mCherry intensities at the NDC80 centroid locations, $I_{NDC80}$, and INCENP-mCherry intensity at the midpoint between two sister kinetochores, $I_{mid}$, were measured by two-dimensional cubic interpolation with *scipy.interpolate.griddata* function. For each cell, we used $\bar{I}_{mid}$, which is $I_{mid}$ averaged over kinetochores in the images before chemical treatments, and cytoplasmic background level, $I_{bg}$, to obtain normalized the INCENP-mCherry intensities at NDC80, $I_{NDC80}^{norm}$, by:

$$I_{NDC80}^{norm} = \frac{I_{NDC80} - \bar{I}_{mid}}{\bar{I}_{mid} - I_{bg}}$$

Kinetochores with similar K-K distances were grouped in the same way as in *Figures 3D* and *5E*, and then the normalized INCENP-mCherry intensities at NDC80, $I_{NDC80}^{norm}$, were averaged within each group. The normalized INCENP-mCherry intensity was converted to Aurora B concentration in *Figures 7A* by assuming that $\bar{I}_{mid}$ corresponds to the peak Aurora B concentration, which was previously estimated to be 10 µM (*Zaytsev et al., 2016*).

## Drug treatments

Cells were incubated with 5 µM Nocodazole (Sigma Aldrich) for >10 min for microtubule depolymerization. Aurora B inhibition was performed by adding 3 µM of ZM447439 (Enzo Life Sciences) during imaging. Taxol (Enzo Life Sciences) treatment was performed at 10 µM final concentration for >10 min. S-Trityl-L-cysteine (STLC, Sigma Aldrich) treatment was performed at 5 µM final concentration for >60 min to induce monopolar spindles. For the haspin kinase inhibition, cells were treated with 10 µM 5-iodotubercidin (5-ITu, Enzo Life Sciences) for >10 min. The double treatment of 5-ITu and taxol or STLC was performed sequentially by treating cells with 10 µM taxol or 5 µM STLC and then adding 10 µM 5-ITu.

## Mathematical modeling

Here we describe the mathematical model presented in *Figure 7* in detail. The model predicts NDC80 binding fraction from Aurora B concentration at NDC80 in three steps: (1) Aurora B activation dynamics, consisting of autoactivation in trans and deactivation, which determines the concentration of *active* Aurora B from the concentration of Aurora B; (2) NDC80 phosphorylation, which is dependent on the active Aurora B concentration; and (3) NDC80-kMT binding, which is governed by the phosphorylation level of NDC80.

### (1) Aurora B activation

In this section, we present a quantitative model for the relationship between the Aurora B concentration (which we measured in *Figure 6*) and the *active* Aurora B concentration (which determines the steady-state level of NDC80 phosphorylation). It has been previously argued that Aurora B activation is predominately due to active Aurora B phosphorylating inactive Aurora B in trans (*Zaytsev et al., 2016*; *Xu et al., 2010*; *Sessa et al., 2005*; *Bishop and Schumacher, 2002*), which we incorporate into our model. We model Aurora B at the location of NDC80 as consisting of two separate pools: one that is dependent on haspin, and the other that is not. We assume that those two Aurora B pools do not interact with each other, and independently undergo auto-activation in trans. We further assume that the phosphatase activity proceeds at a constant rate for each pool.

We denote the haspin-dependent and haspin-independent pools of Aurora B by $A_{hd}$ and $A_{hi}$, respectively. Then the inter-molecular autoactivation by in trans phosphorylation and inactivation by dephosphorylation for each of the two Aurora B pools are described by:

$$\begin{cases} A_x^{active} + A_x^{inactive} \xrightarrow{k_x^a} A_x^{active} + A_x^{active} \\ A_x^{active} \xrightarrow{k_x^d} A_x^{inactive} \end{cases} \qquad x = hd \text{ or } hi$$

where $A_x^{active}$ and $A_x^{inactive}$ are the active and inactive Aurora B in pool $x$, respectively, and $k_x^a$ and $k_x^d$ are the rates of Aurora B activation and deactivation for the pool $x$, respectively. Thus, an ordinary differential equation (ODE) for active Aurora B concentration can be written as:

$$\begin{aligned} \frac{\partial \left[A_x^{active}\right]}{\partial t} &= k_x^a \left[A_x^{active}\right]\left[A_x^{inactive}\right] - k_x^d \left[A_x^{active}\right] \\ &= k_x^a \left[A_x^{active}\right]\left(\left[A_x\right] - \left[A_x^{active}\right]\right) - k_x^d \left[A_x^{active}\right] \\ &= k_x^a \left[A_x^{active}\right]\left(\left[A_x\right] - \frac{k_x^d}{k_x^a} - \left[A_x^{active}\right]\right) \end{aligned}$$

where $[A_x] = \left[A_x^{active}\right] + \left[A_x^{inactive}\right]$ is the concentration of the pool $x$. The steady-state solution for this ODE is:

$$\left[A_x^{active}\right] = \begin{cases} 0, & [A] < K_x \\ [A_x] - K_x, & [A] \geq K_x \end{cases}$$

where $K_x \equiv k_x^d / k_x^a$ is the equilibrium constant for the Aurora B activation for pool $x$. We can infer that $\left[A_{hi}^{active}\right] = [A_{hi}] - K_{hi}$ is positive, because Aurora B still acts on NDC80 after the removal of the haspin-dependent pool (*Figure 5*). Therefore, the total concentration of active Aurora B at NDC80 can be written as:

$$
\begin{aligned}
[A^{active}] \;&=\; [A_{hd}^{active}] + [A_{hi}^{active}] \\
&=\;
\begin{cases}
[A_{hi}] - K_{hi}, & [A] < K_{hd} + [A_{hi}] \\
[A] - K_{hd} - K_{hi}, & [A] \geq K_{hd} + [A_{hi}]
\end{cases} \\
&=\;
\begin{cases}
A^{*} - K, & [A] < A^{*} \\
[A] - K, & [A] \geq A^{*}
\end{cases}
\end{aligned}
\tag{1}
$$

where $[A] = [A_{hd}] + [A_{hi}]$ is the total concentration of Aurora B at NDC80, $K = K_{hd} + K_{hi}$, and $A^{*} = K_{hd} + [A_{hi}]$ is a threshold Aurora B concentration, which is the minimum concentration of Aurora B required for the activity of Aurora B to increase with its concentration.

## (2) NDC80 phosphorylation

In this section, we present a mathematical model to relate the total concentration of active Aurora B at NDC80, $[A^{active}]$, to the phosphorylation level of NDC80. Active Aurora B may phosphorylate multiple Aurora B phosphorylation sites in each Ndc80 N-terminal tail (*Guimaraes et al., 2008*), which we describe with the equations:

$$
\begin{cases}
A^{active} + (\text{dephosphorylated site}) \xrightarrow{k_p} A^{active} + (\text{phosphorylated site}) \\
(\text{phosphorylated site}) \xrightarrow{k_{dp}} (\text{dephosphorylated site})
\end{cases}
$$

The corresponding ODE for the number of phosphorylated sites is:

$$
\begin{aligned}
\frac{\partial N_p}{\partial t} &= k_p[A^{active}]N_{dp} - k_{dp}N_p \\
&= k_p[A^{active}](N - N_p) - k_{dp}N_p
\end{aligned}
$$

where $N_p$, $N_{dp}$, and $N = N_p + N_{dp}$ is the number of phosphorylated sites, dephosphorylated sites, and the total number of sites per kinetochore, respectively. The steady-state solution for the ODE gives:

$$
f_{phos} = \frac{N_p}{N} = \left(1 + \frac{K_{phos}}{[A^{active}]}\right)^{-1}
\tag{2}
$$

where $f_{phos}$ is the fraction of phosphorylated sites, and $K_{phos} \equiv \frac{k_{dp}}{k_p}$ is the equilibrium constant for NDC80 phosphorylation. Plugging *Equation 1* into *Equation 2* yields:

$$
f_{phos} =
\begin{cases}
\left(1 + \frac{K_{phos}}{A^{*} - K}\right)^{-1}, & [A] < A^{*} \\
\left(1 + \frac{K_{phos}}{[A] - K}\right)^{1}, & [A] \geq A^{*}
\end{cases}
\tag{3}
$$

## (3) NDC80 binding

In this section, we present a model to relate the fraction of phosphorylated sites in NDC80 per kinetochore, $f_{phos}$, to the fraction of NDC80 bound to kMTs (which we measure using FLIM-FRET). Assuming that the number of available binding sites for NDC80 is constant, we may describe the NDC80 binding and unbinding by the following equations:

$$
\begin{cases}
(NDC80\ unbound) \xrightarrow{k_{on}} (NDC80\ bound) \\
(NDC80\ bound) \xrightarrow{k_{off}} (NDC80\ unbound)
\end{cases}
$$

The corresponding ODE for the number of NDC80 bound to kMTs is:

$$
\begin{aligned}
\frac{\partial n_{on}}{\partial t} &= k_{on}n_{off} - k_{off}n_{on} \\
&= k_{on}(n - n_{on}) - k_{off}n_{on}
\end{aligned}
$$

where $n_{on}$ and $n_{off}$ are the number of NDC80 bound and unbound to kMTs, respectively, and $n = n_{on} + n_{off}$ the total number of NDC80 per kinetochore. Solving for the steady state gives:

$$
f_{bound} \equiv \frac{n_{on}}{n} = (1 + K_{binding})^{-1}
\tag{4}
$$

where $K_{binding} = k_{off}/k_{on}$ is the equilibrium constant for the NDC80 binding, and $f_{bound}$ is the NDC80 binding fraction.

The binding affinity of NDC80 decreases with the phosphorylation level of NDC80 (*Figure 4*) (*Zaytsev et al., 2015*), arguing that $K_{binding}$ is a function of $f_{phos}$. Since $f_{phos}$ is small in late prometaphase and metaphase ($< 1/9$), we approximate the function by a first-order polynomial, that is, $K_{binding}(f_{phos}) \approx K_0 + K_0' f_{phos}$, and consequently *Equation 4* becomes:

$$f_{bound} = \left(1 + K_0 + K_0' f_{phos}\right)^{-1} \tag{5}$$

Combining *Equations 3 and 5*, we have the relationship between the total Aurora B concentration $[A]$ and the NDC80 binding fraction $f_{bound}$ as:

$$f_{bound} = \begin{cases} \left(1 + K_0 + K_0'\left(1 + \frac{K_{phos}}{A^* - K}\right)^{-1}\right)^{-1}, & [A] < A^* \\ \left(1 + K_0 + K_0'\left(1 + \frac{K_{phos}}{[A] - K}\right)^{-1}\right)^{-1}, & [A] \geq A^* \end{cases} \tag{6}$$

We first determined the parameters $K_0$ and $K_0'$ by fitting *Equation 5* to the NDC80 binding fraction vs. phosphorylation level data in *Figure 4F*, which yielded $K_0 = 1.43 \pm 0.06$ (SE) and $K_0' = 18 \pm 2$ (SE). To estimate the remaining three free parameters, $K$, $K_{phos}$, and $A^*$, we fit *Equation 6* to the NDC80 binding fraction vs. Aurora B concentration at NDC80 data (*Figure 7*), and obtained $K = 3.5 \pm 0.4\ \mu M$ (SE), $K_{phos} = 19 \pm 5\ \mu M$ (SE), and $A^* = 4.6 \pm 0.2\ \mu M$ (SE).

## Supplemental experiments
### Measurement of the fraction of β-tubulin labeled with TC-FlAsH
To measure the fraction of β-tubulin labeled with TC-FlAsH, we sought to determine the concentration of labeled β-tubulin in the cell, and divide it by the total concentration of β-tubulin. We calculated the concentration of labeled β-tubulin by combining 3D fluorescence microscopy to measure the total fluorescence of β-tubulin-TC-FlAsH per cell, and fluorescence correlation spectroscopy (FCS) to measure the fluorescence per molecule of TC-FlAsH.

### 3D fluorescence microscopy
We acquired z-stacks of β-tubulin-TC-FlAsH in mitotic cells using two-photon fluorescence microscopy (*Figure 1—figure supplement 1A*), and then segmented the 3D images using an active contour approach (*Figure 1—figure supplement 1B*). Assuming that the cytoplasmic background results from FlAsH binding specifically to monomeric β-tubulin and nonspecifically to cysteine-rich proteins freely diffusing in the cytoplasm, the average number of photons emitted from β-tubulin-TC-FlAsH in microtubules is the difference between the average photon rate throughout the entire cell ($423 \pm 33$ ms$^{-1}$) and the average photon rate in the cytoplasm (determined from the mode of fluorescence distribution within each segmented image, $327 \pm 30$ ms$^{-1}$), which is $96 \pm 12$ ms$^{-1}$ (*Figure 1—figure supplement 1C*). The instrumental setting of two-photon fluorescence microscopy was identical to that of the FLIM system described above, where the imaging parameters are: laser wavelength, 865 nm; excitation intensity, 3 mW; integration time, 3 s; z-stack separation, 0.5 μm; scanning area, 27.5 × 27.5 μm.

### Fluorescence correlation spectroscopy
To convert the measured photon rate from fluorescence microscopy to a measurement of the absolute concentration of β-tubulin-TC-FlAsH, we used two-photon FCS to determine the volume of the point spread function (PSF) and the molecular brightness (i.e. the number of photons emitted per molecule per second) of TC-FlAsH (*Hess and Webb, 2002*).

First, we performed an FCS measurement on 97 nM Alexa Fluor 488 (Thermo Fisher) in water. FCS measurements were performed on the same instrumental setting as the 3D fluorescence microscopy described above, with laser intensity 5 mW. five autocorrelation functions, each of which had been collected for 300 s, were averaged, and then the following FCS model, $G_D(\tau)$, was fitted to the average autocorrelation function:

$$G_D(\tau) = \frac{1}{V_{eff}\chi^2 C}\left(\frac{1}{1+8D\tau/w_{xy}^2}\right)\left(\frac{1}{1+8D\tau/w_z^2}\right)^{\frac{1}{2}} + G_\infty$$

where $V_{eff}$ is the effective volume of PSF, $C$ the concentration of fluorophores (which is 97 nM), $\chi^2$ the background noise correction factor (*Hess and Webb, 2002*), $D$ the diffusion coefficient of Alexa Fluor 488, which was previously estimated to be 435 μm$^2$/s (*Petrásek and Schwille, 2008*), and $w_{xy}$ and $w_z$ are the radial and axial beam waists, respectively (*Figure 1—figure supplement 1D*). $w_z$ can be written in terms of $V_{eff}$ and $w_{xy}$:

$$w_z = \left(\frac{2}{\pi}\right)^{3/2}\frac{V_{eff}}{w_{xy}^2}$$

Fitting the FCS model to the Alexa 488 FCS data estimated $V_{eff}$ and $w_{xy}$ to be 0.364 ± 0.004 μm$^3$ and 278 ± 4 nm, respectively (*Figure 1—figure supplement 1D*).

We next performed an FCS measurement on a synthesized TC peptide labeled with FlAsH. 50 μM synthesized TC peptide (Ac-AEEEACCPGCC-NH$_2$, Genscript), 100 μM FlAsH-EDT$_2$, and 10 mM 2-mercaptoethanesulfonate (Sigma Aldrich) were incubated for an hour to associate TC peptide with FlAsH, then diluted in the imaging buffer by 500 times, and prepared on a coverslip for FCS measurement. The laser intensity was set to 3 mW. six autocorrelation functions, each of which had been collected for 300 s, were averaged, and the following FCS model was fitted to the average autocorrelation function to determine the number of fluorophores $N$ in a focal volume $V_{eff}$:

$$G_D(\tau) = \frac{1}{N\chi^2}\left(\frac{1}{1+\tau/\tau_D}\right)\left(\frac{1}{1+\left(w_{xy}^2/w_z^2\right)(\tau/\tau_D)}\right)^{\frac{1}{2}} + G_\infty$$

while $w_{xy}$ and $w_z$ were fixed to the values determined from the FCS measurement on Alexa Fluor 488 (*Figure 1—figure supplement 1E*). The photon count collected during the FCS measurement was corrected for background noise, and then divided by $N$ to yield the molecular brightness of TC-FlAsH, 233.4 ± 9.3 s$^{-1}$. Using the estimated molecular brightness and the effective volume of the PSF, we calculated the average concentration of the polymerized β-tubulin-TC-FlAsH to be

$$\frac{9.6\times10^4\ s^{-1}}{(233.4\ s^{-1})(0.364\ \mu m^3)} = 1.13\times10^{21}m^{-3} = 1.88\pm0.13\ \mu M$$

## Calculating labeling ratio

A previous study (*Dumontet et al., 1996*) estimated the percentage of polymerized β-tubulin in a mitotic human tissue culture cell to be 36 ± 7%. Combining this information with our estimate of an average concentration of polymerized β-tubulin-TC-FlAsH of 1.88 ± 0.13 μM leads to a total concentration of β-tubulin-TC-FlAsH of 1.88 μM × 100/36 ≈ 5.22 ± 1.08 μM. Since the total concentration of tubulin dimer in a tissue culture cell is ~ 20 μM (*Hiller and Weber, 1978*), we estimated the fraction of labeled β-tubulin to be 5.22 μM/20 μM ≈ 26.1 ± 5.4%. This estimate makes use of data obtained in different cell types, which may introduce inaccuracies. Systematic errors in the total concentration of tubulin or the concentration of the β-tubulin-TC-FlAsH will produce proportional errors in the estimates of the fraction of labeled β-tubulin, resulting in proportional systematic errors in the conversion of NDC80 FRET fraction to NDC80 binding fraction.

## Förster radius estimation

To measure the Förster radius $R_0$ of FRET between mTurquiose2 and TC-FlAsH, we created a construct containing mTurquoise2 tethered to TC (mTurquoise2-TC), expressed it in U2OS cells, and acquired fluorescence decays of mTurquoise2 using FLIM, which were well-described by a single-exponential fluorescence decay with a lifetime of 3.75 ± 0.03 ns (SD) in the absence of FlAsH labeling (*Figure 1—figure supplement 4A*). When FlAsH is added to these cells, FLIM measurements revealed the presence of additional shorter-lifetime species, corresponding to mTurquoise2 molecules engaged in FRET with TC-FlAsH (*Figure 1—figure supplement 4A*). Then we performed Monte Carlo protein simulations (which are described below) to model the conformational ensemble

of the flexible tether between mTurquoise2 and TC-FlAsH and obtain the distribution $p(r)$ of the distance $r$ between mTurquoise2 and FlAsH (Figure 1–figure supplement 4B). The fluorescence lifetime $\tau$ of donors engaged in FRET is related to the donor-acceptor distance $r$ by:

$$\tau(r; R_0) = \frac{\tau_D}{1 + \left(\frac{R_0}{r}\right)^6}$$

where $\tau_D$R0 is the fluorescence lifetime of the donor in non-FRET state ($3.75 \pm 0.03$ ns), and $R_0$ is the Förster radius. Therefore, the fluorescence decay $y(t)$R0 of mTurquoise2-TC-FlAsH can be modeled as:

$$y(t) = A_\mathrm{D} \exp\left(-\frac{t}{\tau_D}\right) + A_\mathrm{FRET} \int p(r) \exp\left(-\frac{t}{\tau(r; R_0)}\right) dr$$

where $A_\mathrm{D}$ is the population in the non-FRET state and $A_\mathrm{FRET}$ is that in the FRET state, both of which are free parameters of the model along with $R_0$. This model (after convolved with the IRF) was fit to the measured fluorescence decay curve of mTurquoise2-TC-FlAsH, allowing us to estimate the Förster radius $R_0$ to be $5.90 \pm 0.10$ nm (**Figure 1—figure supplement 4C**).

## Characterization and calibration of NDC80-kMT FLIM-FRET measurements by Monte Carlo simulations

### Characterization of NDC80-kMT FRET vs NDC80-MT distance relationship

To characterize the FRET between mTurquoise2-NDC80 and FlAsH-labeled microtubule when NDC80 is not bound to the microtubule, we performed large-scale, atomistic Monte Carlo protein simulations to model the conformational ensemble of the tether between mTurquoise2 and NDC80 and the disordered C-terminal tails of twelve β-tubulins near the NDC80 complex (which is described below) (**Figure 1—figure supplement 5A**). 4000 sets of positions of mTurquoise2 and TC were generated for each case where the NDC80 bound to an inter- or intra-tubulin dimer interface was translated away from the microtubule by a certain distance (0~15, 0.5 nm increment) in a direction perpendicular to the microtubule surface. For each randomly sampled set of distances between mTurquoise2 and TC-FlAsH, $\vec{r} = \{r_i\}$, the fluorescence lifetime was calculated by:

$$\tau\left(\vec{r}\right) = \frac{\tau_D}{1 + \sum_{i=1}^{12} I_i \left(\frac{R_0}{r_i}\right)^6}$$

where $I_i \sim \mathrm{Bernoulli}(f_{label})$ indicates whether or not the $i$-th TC motif is labeled with FlAsH-EDT$_2$; $R_0$ is the Förster radius between mTurquoise2 and TC-FlAsH; and $\tau_D$=3.75 ns is the non-FRET lifetime of mTurquoise2 (**Figure 1—figure supplement 5A**). The estimated labeling fraction 26.1% and the Förster radius 5.90 nm were used for $f_{label}$ and $R_0$, respectively. For each NDC80-kMT distance, 2 million fluorescence lifetimes were sampled, based on which we simulated 30 fluorescence decay curves of mTurquoise2 by:

$$\mathrm{Poisson}\left(A \int p\left(\vec{r}\right) \exp\left(-\frac{t}{\tau\left(\vec{r}\right)}\right) d\vec{r}\right)$$

where the amplitude $A$ of the fluorescence decay was set to 5000 (**Figure 1—figure supplement 5A**). Single- and double-exponential decay models were fit to the simulated fluorescence decays (by maximum likelihood method) (**Figure 1—figure supplement 5B**, bottom). The Bayesian information criterion (BIC) was used as a criterion for model selection between the single- and double-exponential decay models. The difference in BIC between single- and double-exponential models, $\Delta\mathrm{BIC} = \mathrm{BIC}_{1\mathrm{expo}} - \mathrm{BIC}_{2\mathrm{expo}}$, was plotted against the NDC80-kMT distance (**Figure 1—figure supplement 5B**, top right). $\Delta\mathrm{BIC}$ is negative when NDC80-kMT distance is larger than 8 nm, indicating that single-exponential model performs better than double-exponential model in terms of the goodness of fit and the complexity of model (**Figure 1—figure supplement 5B**).

## NDC80-kMT FRET fraction calibration

To obtain the relationship between NDC80 FRET fraction and NDC80 binding fraction, we performed large-scale Monte Carlo simulations to obtain 4000 sets of distances between mTurquoise2 and TC-FlAsH, $\vec{r} = \{r_i\}$, for each case where mTurquoise2-NDC80 is bound to the TC-tagged microtubule at the inter- or intra-dimer interface. Then we sampled 0.5 million fluorescence lifetimes $\tau$ as described above, and simulated a fluorescence decay curve for the situation where a fraction $f_b$ of mTurquoise2-NDC80 are bound to microtubules and have lifetime $\tau\left(\vec{r}\right)$, while the other $1 - f_b$ are not bound and have lifetime $\tau_D$ (**Figure 1—figure supplement 5C**, green dots):

$$\mathrm{Poisson}\left(A\left[(1-f_b)\exp\left(-\frac{t}{\tau_D}\right) + f_b\int p\left(\vec{r}\right)\exp\left(-\frac{t}{\tau\left(\vec{r}\right)}\right)d\vec{r}\right]\right)$$

where the amplitude $A$ of the fluorescence decay curve was set to 10,000 (**Figure 1—figure supplement 5C**, left). The simulated fluorescence decay curve was then fit by two-exponential fluorescence decay model (**Figure 1—figure supplement 5C**, red lines):

$$A'\left[(1-f_{FRET})\exp\left(-\frac{t}{\tau_D}\right) + f_{FRET}\exp\left(-\frac{t}{\tau_{FRET}}\right)\right]$$

to acquire the FRET fraction, $f_{FRET}$. The data of NDC80 FRET fraction, $f_{FRET}$, vs NDC80 binding fraction, $f_b$, was fit using a linear model $f_{FRET} = af_b$ (**Figure 1—figure supplement 5C**, right). To determine the uncertainty in the slope $a$ (gray area in **Figure 1—figure supplement 5C**, right), we repeated the process above with the mean ± error values of $R_0$ and beta-tubulin labeling fraction $f_{label}$. As a result, we obtained $a = 0.42 \pm 0.08$, and used this calibration to convert NDC80 FRET fraction to NDC80 binding fraction.

## Monte Carlo protein simulations

Atomistic simulations were performed by the CAMPARI (v2) package (**Vitalis and Pappu, 2009b**), employing the ABSINTH implicit solvation model and forcefield paradigm (**Vitalis and Pappu, 2009a**) at the intrinsic solvation (IS) limit (**Das and Pappu, 2013**) (unless stated otherwise), where the energy function is simply a combination of Lennard-Jones energy and ABSINTH solvation energy. *mTurquoise2-TC construct simulation*: For the Förster radius estimation, we ran 50 independent simulations on the mTurquoise2-TC construct in spherical soft-wall boundary conditions with radius 100 Å. An input structure was used only for the folded mTurquoise2 domain (adapted from PDB 4B5Y), and we employed CAMPARI to generate the tether and TC domains (GMDELYKYSDLF LNCCPGCCMEP) from scratch. To prevent unphysical unfolding and/or conformational change of mTurquoise2, we imposed constraints on internal degrees of freedom of residues in the folded region. Each simulation consisted of $2 \times 10^6$ MC steps with sampling frequency of $(5000\ \mathrm{steps})^{-1}$, and the simulation temperature was set to 400 K to scan a large structural ensemble. Note that the system is quickly relaxed in the intrinsic solvation (IS) limit, and hence it does not require a long simulation time to reach equilibration. The average coordinate of the alpha carbons of 4 residues before and after Trp66 was used as the location of mTurquoise2 chromophore, and the average coordinate of alpha carbons of the four cysteine residues in the TC motif was used as the location of TC-FlAsH.

## mTurquoise2-NDC80 and TC-labeled microtubule simulation

We constructed a system consisting of the NDC80 complex, 12 tubulin dimers as described in **Figure 1—figure supplement 5A**. The initial structures of the system were constructed by combining the structures of microtubule (PDB 3JAS) (**Zhang et al., 2015**) and one of bonsai-NDC80s attached to a tubulin dimer at the inter- and intra-dimer interfaces (PDB 3IZ0) (**Alushin et al., 2010**), and then incorporating the structures of disordered regions (tether, GMDEL YKYSD LMET, and C-terminal tail +TC, SEYQQ YQDAT AEEEE DFGEE AEEEA CCPGC C) generated by the loop modeling module of Rosetta 3.8 (**Mandell et al., 2009**; **Leaver-Fay et al., 2011**). Clashes in the initial structure were removed by the Rosetta relaxation module (**Nivón et al., 2013**). For folded regions (where we have structure information from PDB), we imposed constraints on internal degrees of freedom as before. To prevent dissociation of microtubule into individual tubulins, we also imposed a harmonic restraint

potential on atoms at the interface of two different chains. If two atoms from different chains are closer than 20 Å, the pair contributes an additional potential

$$E_{\text{drest}}(i,j) = k\left(r_{ij} - r_{ij}^{0}\right)^{2},$$

where $i$ and $j$ are atomic indices, $r_{ij}$ is the distance between two atoms $i$ and $j$, $r_{ij}^{0}$ is the initial distance between $i$ and $j$, and $k$ is a force constant (set to 3.0 kcal/mol/Å$^2$). We employed spherical soft-wall boundary conditions with radius 200 Å, and the simulation temperature of 400 K. We ran a relaxation simulation for $2 \times 10^6$ MC steps, and a production simulation for another $2 \times 10^6$ MC steps with sampling frequency of $(500 \text{ steps})^{-1}$. For the FRET efficiency vs NDC80-MT data (*Figure 1—figure supplement 5B*), we used the final structure of the relaxation simulation and translated the NDC80 complex along the axis orthogonal to the microtubule surface by several distance values: from 0 to 15 nm by increment of 0.5 nm. For each system, a simulation of $2 \times 10^6$ MC steps was conducted to generate data with sampling frequency of $(500 \text{ steps})^{-1}$.

### Protein structure illustration

Protein structure illustrations were generated by The PyMOL Molecular Graphics System, Version 2.0 Schrödinger, LLC.

### Quantification and statistical analysis

The statistical test used, sample size (number of cells and kinetochores), dispersion and precision measures can be found in figure legends, Results, or below. All curve fittings, except FLIM data analysis (which is separately explained above), were performed by Levenberg-Marquardt algorithm with residuals weighted by the inverse of y-errors, and the corresponding 95% confidence intervals were calculated by *predint* function in MATLAB. To assess the significance of correlation, we determined p-value from $1 - \alpha$, where $\alpha$ is the smallest confidence level that makes zero contained in the confidence interval of the slope of the linear fit.

## Acknowledgements

We thank A Murray and N Kleckner for comments on the manuscript; J DeLuca, M Lampson, and I Cheeseman for reagents; F Rago for help with retroviral transfection; D Kim for help with nucleofection; Needleman lab members and J Oh for proof reading, comments and discussion.

## Additional information

### Funding

| Funder | Grant reference number | Author |
| --- | --- | --- |
| National Science Foundation | DBI-0959721 | Daniel J Needleman |
| National Institutes of Health | R01NS056114 | Rohit V Pappu |
| National Science Foundation | DMR-0820484 | Daniel J Needleman |

The funders had no role in study design, data collection and interpretation, or the decision to submit the work for publication.

### Author contributions

Tae Yeon Yoo, Conceptualization, Data curation, Software, Formal analysis, Validation, Investigation, Visualization, Methodology, Writing—original draft, Writing—review and editing; Jeong-Mo Choi, Software, Formal analysis, Investigation, Writing—review and editing; William Conway, Formal analysis, Investigation, Writing—review and editing; Che-Hang Yu, Methodology, Built, optimized and maintained the FLIM instrument used in this study, Provided input for applying the FLIM technique to the measurement of NDC80 binding; Rohit V Pappu, Supervision, Funding acquisition, Writing—review and editing; Daniel J Needleman, Conceptualization, Supervision, Funding acquisition, Writing—original draft, Writing—review and editing

## Author ORCIDs
Tae Yeon Yoo (iD) http://orcid.org/0000-0002-8145-1051
Jeong-Mo Choi (iD) http://orcid.org/0000-0003-2656-4851
Che-Hang Yu (iD) https://orcid.org/0000-0002-0353-9752
Rohit V Pappu (iD) http://orcid.org/0000-0003-2568-1378

## Decision letter and Author response
Decision letter https://doi.org/10.7554/eLife.36392.024
Author response https://doi.org/10.7554/eLife.36392.025

## Additional files

### Supplementary files
• Transparent reporting form
DOI: https://doi.org/10.7554/eLife.36392.020

### Data availability
All microscopy image data and data points in the presented plots have been deposited in Dryad (DOI: https://doi.org/10.5061/dryad.14rr125). Analysis codes are deposited in GitHub; DOIs provided in the manuscript.

The following dataset was generated:

| Author(s) | Year | Dataset title | Dataset URL | Database, license, and accessibility information |
|---|---|---|---|---|
| Yoo TY, Choi J, Conway W, Yu C, Pappu RV, Needleman DJ | 2018 | Data from: Measuring NDC80 binding reveals the molecular basis of tension-dependent kinetochore-microtubule attachments. | https://dx.doi.org/10.5061/dryad.14rr125 | Available at Dryad Digital Repository under a CC0 Public Domain Dedication |

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
