## [Decision Letter]

Thank you for submitting your article "Measuring NDC80 binding reveals the molecular basis of tension-dependent kinetochore-microtubule attachments" for consideration by *eLife*. Your article has been reviewed by three peer reviewers, and the evaluation has been overseen by a Reviewing Editor and Anna Akhmanova as the Senior Editor. The reviewers have opted to remain anonymous.

The reviewers have discussed the reviews with one another and the Reviewing Editor has drafted this decision to help you prepare a revised submission.

General assessment:

This is a well-written and clearly presented manuscript describing a powerful assay to quantitatively measure the interaction between the kinetochore and kinetochore microtubules and address a central question about the molecular mechanism of error correction at kinetochores. This manuscript provides the first glimpse at the dynamic nature of the interaction between the NDC80 complex, the principle microtubule-binding interface of the eukaryotic kinetochore, and the microtubule lattice. The authors develop a sophisticated data acquisition and analysis system to obtain high quality FLIM-FRET information about the proximity of the NDC80 complex to the tubulin lattice.

Central conclusions:

Using this system, the authors make a number of novel observations:

1) They propose that ~ 35% of the Ndc80 molecules in a metaphase kinetochore are bound to the microtubule lattice at steady state.

2) The bound fraction of Ndc80 molecules increases from prometaphase to metaphase. The authors propose that the observed increase reflects chromosome-autonomous error correction processes.

3) The authors also find that the Ndc80 bound fraction correlates with centromere tension, and that this correlation requires the activity of the centromere-bound Aurora B kinase.

4) Finally, the authors propose a simple mathematical model (using a previously established model from the Grishchuck lab) to integrate their observations into a mechanistic framework.

The authors are to be commended on developing a state of the art FLIM-FRET assay and also a large-scale simulation approach in order to maximize their ability to obtain biologically relevant insight in terms of the bound fraction of Ndc80 molecules. The extensive efforts represent really the first measurement of this critical parameter.

Altogether, the work is technically impressive and an important contribution to our understanding of the architecture and dynamics of the mammalian kinetochore-microtubule interface. The paper is well written and the data overall is presented clearly, and in particular the authors do a good job relating their measurements to other ones in the field.

Essential revisions:

The reviewers raise a number of concerns that must be adequately addressed before the paper can be accepted. Some of the required revisions likely require further experimentation within the framework of the presented studies and techniques.

1) What is a "bound" versus "unbound" NDC80 complex? The difficulty in classification arises because Ndc80 can bind to the microtubule via its CH-domain as well as the 80 amino acid long unstructured tail. The authors monitor the binding of the CH-domains, but not that of the N-terminal tail. This is problematic, because the unstructured tail contributes to microtubule binding both in vitro and in vivo as shown by studies from Sophie Dumont's lab (Long et al., Current Biology 2017). It is reasonable to expect that the tail can extend to lengths of 5-10 nm while maintaining attachment to the lattice. This would place the CH-domain out of FRET range with the tubulin lattice even as the Ndc80 molecule is bound. The authors might have to address this ambiguity via both experimentation and simulation. Specifically, they could use phosphomutants of Ndc80 to explicitly define the FRET signature of unbound and bound Ndc80 molecules.

2) Is the Ndc80 bound fraction on a per kinetochore basis or a per kMT basis? This is a significant issue that is not discussed; it will nonetheless have significant bearing on the interpretation of the data. The calibration simulations used here are based on Ndc80 molecules binding to the microtubule lattice. But the experiments measure Ndc80 binding on a per kinetochore basis. This distinction is important to consider. Given a microtubule in its proximity, the binding of Ndc80 to the lattice is subject only to phosphoregulation (this is the implicit focus of the manuscript). However, the measured fraction of Ndc80's per kinetochore is also dependent on how many microtubules are attached. If the number of microtubules per kinetochore increases (this is known to happen from prometaphase to metaphase in other model systems), then the bound fraction of Ndc80 molecules will increase even in the absence of any chromosome-autonomous regulation of Ndc80 affinity. Without studying whether and how the number of kinetochore-bound microtubules changes, the authors cannot conclude that the bound-fraction increase is the result of an active regulatory mechanism.

Along the same lines, it is known that the human kinetochore binds ~ 20 microtubules in metaphase, but it has the capacity to bind ~ 25 microtubules. Therefore, it will contain a fraction of Ndc80 molecules that are always unbound. This will introduce an offset in their discussions of the bound fraction. The authors should note this in their Discussion.

3) Figure 2: One possible contribution to the variability of the FRET ratio for the "off-centered" pairs are that kinetochores in this population may have different attachment geometries (lateral vs. end-on), or be regulated by different biochemical systems (e.g. AurA) in different locations. The authors should comment on these possibilities as they can impact the "chromosome autonomous" conclusion.

4) In Figure 2D, it appears that for "centered" kinetochores, the NDC80 FRET fraction continuously increases throughout mitosis. However, it seems to me that NDC80 itself is required to generate centromere tension, by linking the dynamic microtubule plus-ends to the centromere. Therefore, does this imply that average kinetochore-kinetochore distances would also tend to increase continuously as more NDC80 molecules bind, perhaps recruiting additional kinetochore microtubules? If so, is this observed experimentally? If not, it would be useful to explain what effect a continuous increase in NDC80 binding would have on the centered sister kinetochore pairs, which would not affect the off-centered pairs.

5) Similarly, in Figure 3D, could the linear increase of NDC80 fraction with increasing K-K distance reflect a dependency of K-K distance on NDC80 fraction – e.g., as more NDC80 molecules bind, the K-K distance tends to be larger because additional kinetochore microtubules contribute pulling forces? The taxol-treated cells have reduced K-K distance and reduced NDC80 FRET fractions, but this result may be difficult to interpret since Taxol suppresses microtubule plus-end dynamics, and may change the flexural rigidity of the microtubules themselves, and so may alter NDC80 binding by itself. Thus, I worry that the conclusion that "tension is a primary regulator of NDC80-kMT binding during error correction" is perhaps not well established by the data in Figure 3D. It would be ideal if K-K distance could be disrupted without altering kinetochore microtubule plus-end dynamics (e.g., perhaps at the minus-ends?), and then data collected on NDC80 FRET fraction.

6) Centromeric tension and lagging versus leading kinetochores: This is an interesting and counterintuitive aspect of the data that went unremarked. The authors show that the bound fraction is a function of tension as well as the status of the kinetochore as lagging versus leading. This observation is counter-intuitive, because a pair of sister kinetochores should be under the same centromeric tension, but one of them is lagging while the other one leading for the vast majority of times. The observation likely hints at two different tension-bearing/force generating interfaces in the kinetochore, which was probably alluded to by the Sophie Dumont study from several years ago. It is important, because the presence of two load-bearing attachments forces one to think of the phosphoregulation of both.

7) Figure 5 and related text: The authors show that haspin inhibition abolishes the relationship between FRET ratio and K-K distance (centromere tension) without affecting steady-state levels of NDC80 binding. It would be stronger if they could specifically rescue tension-dependent in a 5-ITu inhibited context by force-localizing Aurora B to the centromere, if possible to do in a reasonable time frame. As a related point, in general this figure and section of the paper still do not directly show that Aurora B is directly responsible for tension dependence. Are there Aurora B inhibition data points across at least a few different amounts of centromere tension that could be plotted on Figure 5C to show this? It may help to move Figure 5—figure supplement 1 into the main figure since a lot of the interpretation in the text hinges on the comparison between 5-ITu and ZM treatment on the Ndc80 FRET ratios.

---

## [Author Response]

General assessment:This is a well-written and clearly presented manuscript describing a powerful assay to quantitatively measure the interaction between the kinetochore and kinetochore microtubules and address a central question about the molecular mechanism of error correction at kinetochores. This manuscript provides the first glimpse at the dynamic nature of the interaction between the NDC80 complex, the principle microtubule-binding interface of the eukaryotic kinetochore, and the microtubule lattice. The authors develop a sophisticated data acquisition and analysis system to obtain high quality FLIM-FRET information about the proximity of the NDC80 complex to the tubulin lattice.Central conclusions:Using this system, the authors make a number of novel observations:1) They propose that ~ 35% of the Ndc80 molecules in a metaphase kinetochore are bound to the microtubule lattice at steady state.2) The bound fraction of Ndc80 molecules increases from prometaphase to metaphase. The authors propose that the observed increase reflects chromosome-autonomous error correction processes.3) The authors also find that the Ndc80 bound fraction correlates with centromere tension, and that this correlation requires the activity of the centromere-bound Aurora B kinase.4) Finally, the authors propose a simple mathematical model (using a previously established model from the Grishchuck lab) to integrate their observations into a mechanistic framework.The authors are to be commended on developing a state of the art FLIM-FRET assay and also a large-scale simulation approach in order to maximize their ability to obtain biologically relevant insight in terms of the bound fraction of Ndc80 molecules. The extensive efforts represent really the first measurement of this critical parameter.Altogether, the work is technically impressive and an important contribution to our understanding of the architecture and dynamics of the mammalian kinetochore-microtubule interface. The paper is well written and the data overall is presented clearly, and in particular the authors do a good job relating their measurements to other ones in the field.

We would like to thank the editor and the reviewers for their positive comments on our manuscript. We hope that the presented FLIM-FRET technique will also be useful in future studies of kinetochoremicrotubule interactions and regulation. We believe that our simple mathematical model, which builds off of the Grishchuk lab’s prior work, but is a new formalism based on new premises, provides a helpful framework for conceptualizing the regulation of kinetochore-microtubule attachments.

The editor and the reviewers have provided very helpful comments and suggestions. We believe that taking these into account has greatly improved our manuscript. In addition to more minor changes discussed below, we included the following new experiments:

1. Eg5 inhibition: In the original manuscript, we used taxol to study the effect of reducing centromere tension. In the revised manuscript, we also treated cells with the Eg5 inhibitor, STLC, as an alternative way to reduce the tension. This experiment has been performed both in the absence and presence of haspin inhibitor, and these results are added to Figure 3 and 5 of the revised manuscript.

2. Phosphomimetic Hec1 mutants: In the revised manuscript, we used phosphomimetic Hec1 mutants, which have been developed and extensively verified by the DeLuca lab. We replaced the endogenous Hec1/Ndc80 proteins in cells with WT Hec1 or three different phosphomimetic mutants (9A, 2D, or 9DHec1) and repeated some of the experiments we presented in the original manuscript to determine the contributions of Hec1 phosphorylation to the NDC80-kMT binding and to the tension dependency of NDC80-kMT binding. These results are added to Figure 4 and 5 of the revised manuscript.

We believe that these results substantiate the causal relationship between the centromere tension and NDC80-kMT binding, and unveil important aspects of the Aurora B-mediated phosphoregulation of NDC80-kMT binding, as explained below.

Please find the reviewers’ original comments below (in italics) followed by our response to each comment.

Essential revisions:The reviewers raise a number of concerns that must be adequately addressed before the paper can be accepted. Some of the required revisions likely require further experimentation within the framework of the presented studies and techniques.1) What is a "bound" versus "unbound" NDC80 complex? The difficulty in classification arises because Ndc80 can bind to the microtubule via its CH-domain as well as the 80 amino acid long unstructured tail. The authors monitor the binding of the CH-domains, but not that of the N-terminal tail. This is problematic, because the unstructured tail contributes to microtubule binding both in vitro and in vivo as shown by studies from Sophie Dumont's lab (Long et al., Current Biology 2017). It is reasonable to expect that the tail can extend to lengths of 5-10 nm while maintaining attachment to the lattice. This would place the CH-domain out of FRET range with the tubulin lattice even as the Ndc80 molecule is bound. The authors might have to address this ambiguity via both experimentation and simulation. Specifically, they could use phosphomutants of Ndc80 to explicitly define the FRET signature of unbound and bound Ndc80 molecules.

We thank the reviewer for raising the point about the role of the disordered N-terminal tail of Hec1/Ndc80 protein in NDC80 binding and our measurement of NDC80 binding. In our work, the distinction between “bound” and “unbound” NDC80 complex is based on the distance between the CH-domain of Hec1 and microtubules. To make this more explicit, we included the following statement: “Thus, FRET only results when the CH domain of Hec1 is very close to the surface of kMTs, consistent with the short-lifetime species being NDC80 complexes whose Hec1 CH domains are bound to kMTs…”, which is also reiterated in Discussion: “The FLIM-FRET technique developed in this study measures the fraction of NDC80 complexes whose Hec1 CH domains are bound to kMTs.” We focused on the binding of the CH-domain since it is the major microtubule-binding module of the NDC80 complex, as illustrated in previous electron microscopy studies (Alushin et al., 2010). It is certainly an interesting future direction to explore the behavior of the N-terminal tail using FLIM-FRET and protein simulations.

We agree that using phosphomimetic mutants of Hec1 provides valuable information about the contribution of the phosphorylation state of the N-terminal tail on the binding of NDC80 complex. In the revised manuscript, we have included the following additional experiments with phosphomimetic Hec1 mutants: (1) we replaced endogenous Hec1 in cells with WT or three different phosphomimetic Hec1 mutants, 9A-, 2D, and 9D-Hec1, and compared the NDC80 binding (Figure 4B and 4C); (2) we repeated the Aurora B inhibition experiment with 2D-Hec1-expressing cells (Figure 4D); (3) finally, we investigated the NDC80 binding vs K-K distance relationship for NDC80 complex with 9A-Hec1 (Figure 5A and 5B). These experiments revealed that the phosphorylation of the N-terminal tail of Hec1 is the primary mechanism of the Aurora B-mediated regulation of NDC80 binding, and is required for the tension dependency of NDC80-kMT binding.

2) Is the Ndc80 bound fraction on a per kinetochore basis or a per kMT basis? This is a significant issue that is not discussed; it will nonetheless have significant bearing on the interpretation of the data. The calibration simulations used here are based on Ndc80 molecules binding to the microtubule lattice. But the experiments measure Ndc80 binding on a per kinetochore basis. This distinction is important to consider. Given a microtubule in its proximity, the binding of Ndc80 to the lattice is subject only to phosphoregulation (this is the implicit focus of the manuscript). However, the measured fraction of Ndc80's per kinetochore is also dependent on how many microtubules are attached. If the number of microtubules per kinetochore increases (this is known to happen from prometaphase to metaphase in other model systems), then the bound fraction of Ndc80 molecules will increase even in the absence of any chromosome-autonomous regulation of Ndc80 affinity. Without studying whether and how the number of kinetochore-bound microtubules changes, the authors cannot conclude that the bound-fraction increase is the result of an active regulatory mechanism.Along the same lines, it is known that the human kinetochore binds ~ 20 microtubules in metaphase, but it has the capacity to bind ~ 25 microtubules. Therefore, it will contain a fraction of Ndc80 molecules that are always unbound. This will introduce an offset in their discussions of the bound fraction. The authors should note this in their Discussion.

The FLIM-FRET technique developed in this study measures the fraction of NDC80 complexes bound to kMTs. The reviewer raises excellent points that led us to write a new paragraph in Discussion about the contribution of the number of kMTs on NDC80-kMT binding. As stated in the Discussion, we agree that “changes in NDC80 binding fraction might result from either alterations in the binding affinity of NDC80 or variations in the number of kMTs.” However, we argue that “the increase in NDC80 binding with increasing tension is caused by changes in affinity, because: (1) the addition of taxol causes a reduction in tension and a reduction in NDC80 binding, but an increase in the number of kMTs (McEwen et al., 1997); (2) taxol-treated and STLC-treated cells exhibit the same correlation between tension and NDC80 binding (Figure 3D), suggesting that the decrease in binding with decreasing tension occurs independently of the mechanism of perturbation.”Moreover, we estimate based on previous studies that roughly 6 kMTs are sufficient for every NDC80 complex to be within reach of potential binding sites (see the Discussion section for more detail). Therefore NDC80 binding is expected to be insensitive to the number of kMTs if there are at least 6 kMTs, which is far less than the average number kMTs (~20) in human cells. We do think, however, that both the increase in kMT number (observed in McEwen et al., 1997) and the increase in tension (observed in Magidson et al., 2011) contribute to the increase in NDC80 binding in prometaphase, as stated in the same paragraph of the Discussion (third paragraph). In further support of the reviewer’s idea, sister kinetochores in Eg5-inhibited monopolar spindle, which are likely monotelically attached, show differential NDC80-kMT binding (see Figure 3—figure supplement 2, Figure 5—figure supplement 1, and Discussion, third paragraph).

3) Figure 2: One possible contribution to the variability of the FRET ratio for the "off-centered" pairs are that kinetochores in this population may have different attachment geometries (lateral vs. end-on), or be regulated by different biochemical systems (e.g. AurA) in different locations. The authors should comment on these possibilities as they can impact the "chromosome autonomous" conclusion.

We would like to thank the reviewer for pointing out that other mechanisms independent of tension or direct modulation of NDC80 binding affinity may contribute to the increase in the average NDC80 binding during prometaphase and the difference between centered and off-centered kinetochores. To address this comment, we modified the relevant section to include the comment: “NDC80 binding between different subpopulations of kinetochores strongly argues for the existence of chromosome-autonomous regulation, which might be modulated by tension, Aurora kinases A and B, pathways that control the conversion of lateral to end-on kMT attachments, or other factors (Godek et al., 2015, DeLuca et al., 2018)”.

4) In Figure 2D, it appears that for "centered" kinetochores, the NDC80 FRET fraction continuously increases throughout mitosis. However, it seems to me that NDC80 itself is required to generate centromere tension, by linking the dynamic microtubule plus-ends to the centromere. Therefore, does this imply that average kinetochore-kinetochore distances would also tend to increase continuously as more NDC80 molecules bind, perhaps recruiting additional kinetochore microtubules? If so, is this observed experimentally? If not, it would be useful to explain what effect a continuous increase in NDC80 binding would have on the centered sister kinetochore pairs, which would not affect the off-centered pairs.

A previous study (Magidson et al., 2011) measured the K-K distance and position of individual kinetochore pairs throughout the mitosis and showed that K-K distance is small in early prometaphase no matter whether it is centered or off-centered, and gets larger as progressing to the metaphase. Based on this study, we interpret the continuous increase in NDC80 binding of centered kinetochores as follows. Centered kinetochores include (1) kinetochores with attachment errors that are undergoing error correction process but happen to be located at the center and (2) bioriented kinetochores. The fraction of (1) decreases as error correction happens, which leads to the increase in the average tension, which in turn results in the increase in the average NDC80 binding due to the tension dependency we argue for in this manuscript. On the other hand, off-centered kinetochores mostly have erroneous attachment with low tension, and therefore have low NDC80 binding throughout the prometaphase. To address the comment, we added the statement in Discussion: “We hypothesize that the observed increase in NDC80 binding over the course of prometaphase to metaphase results from the combination of changes in kMT number and the changes in NDC80 binding affinity, due to the continual increase in tension at those times (Magidson et al., 2011).”; and also in Results: “We speculate that the temporal increase in NDC80-kMT binding of centered kinetochores is due to the gradual decrease in the number of kinetochores with erroneous attachment that transiently lie on the metaphase plate (Magidson et al., 2011).”

5) Similarly, in Figure 3D, could the linear increase of NDC80 fraction with increasing K-K distance reflect a dependency of K-K distance on NDC80 fraction – e.g., as more NDC80 molecules bind, the K-K distance tends to be larger because additional kinetochore microtubules contribute pulling forces? The taxol-treated cells have reduced K-K distance and reduced NDC80 FRET fractions, but this result may be difficult to interpret since Taxol suppresses microtubule plus-end dynamics, and may change the flexural rigidity of the microtubules themselves, and so may alter NDC80 binding by itself. Thus, I worry that the conclusion that "tension is a primary regulator of NDC80-kMT binding during error correction" is perhaps not well established by the data in Figure 3D. It would be ideal if K-K distance could be disrupted without altering kinetochore microtubule plus-end dynamics (e.g., perhaps at the minus-ends?), and then data collected on NDC80 FRET fraction.

We thank the reviewer for pointing out that Figure 3D alone only shows the correlation, not the causal relationship between NDC80 binding and tension. We have revised the Results section related to Figure 3 so as not to discuss causality at that point in the text.

In order to address the concern pointed out by the reviewer regarding the possible complication in interpreting the results from taxol experiments, we now alternatively reduced tension by inhibiting Eg5 by STLC treatment, which induces the formation of monopolar spindles. In the presence of STLC, K-K distance is significantly decreased (Figure 3E), and poleward-facing kinetochores display a similar correlation between NDC80 binding and tension to taxol-treated cells (Figure 3D), arguing that the correlation is not due to an artifact specific to taxol treatment. In addition, we performed new experiments showing that the correlation between NDC80 binding and K-K distance is eliminated when endogenous Hec1 is replaced with nonphosphorylatable 9A-Hec1 mutant, or when Aurora B is mislocalized by haspin kinase inhibition, even when tension is further reduced using either taxol or STLC (Figure 5A and 5E). This further supports the contention that tension is a primary regulator of NDC80-kMT binding, no matter whether that change in tension occurs during spontaneous oscillations in metaphase or as a result of taxol treatment or STLC treatment.

We believe that the strongest evidence for the causality between NDC80 binding and tension is the results of the haspin inhibition experiments (Figure 5). Hypothetically, the correlation between NDC80 binding and tension could result from either NDC80 binding increasing tension or tension increasing NDC80 binding (or a third factor affecting both). Upon haspin inhibition, the distribution of K-K distances does not change (see Figure 5F), but the correlation between K-K distance and NDC80 binding is eliminated (Figure 5E). This argues that haspin does not affect tension, but rather mediates the connection between tension and NDC80 binding. Furthermore, treating haspin inhibited cells with taxol or STLC causes a reduction in tension, as in cells with active haspin, but this reduced tension is not accompanied by a change in NDC80-kMT binding. This argues that taxol and STLC treatments cause changes in tension, and whether that change in tension is accompanied by a change in NDC80 binding depends on the activity of haspin. Therefore, taken together, we conclude that tension causes changes in NDC80 binding.

6) Centromeric tension and lagging versus leading kinetochores: This is an interesting and counterintuitive aspect of the data that went unremarked. The authors show that the bound fraction is a function of tension as well as the status of the kinetochore as lagging versus leading. This observation is counter-intuitive, because a pair of sister kinetochores should be under the same centromeric tension, but one of them is lagging while the other one leading for the vast majority of times. The observation likely hints at two different tension-bearing/force generating interfaces in the kinetochore, which was probably alluded to by the Sophie Dumont study from several years ago. It is important, because the presence of two load-bearing attachments forces one to think of the phosphoregulation of both.

We agree with the reviewer that the difference between trailing vs. leading kinetochores is an important piece of information for understanding of tension-bearing/force generating interfaces. We have added a sentence to emphasize this and cited Dumont’s paper in the second paragraph of the subsection “NDC80-kMT binding is positively correlated with centromere tension”.

7) Figure 5 and related text: The authors show that haspin inhibition abolishes the relationship between FRET ratio and K-K distance (centromere tension) without affecting steady-state levels of NDC80 binding. It would be stronger if they could specifically rescue tension-dependent in a 5-ITu inhibited context by force-localizing Aurora B to the centromere, if possible to do in a reasonable time frame. As a related point, in general this figure and section of the paper still do not directly show that Aurora B is directly responsible for tension dependence. Are there Aurora B inhibition data points across at least a few different amounts of centromere tension that could be plotted on Figure 5C to show this? It may help to move Figure 5—figure supplement 1 into the main figure since a lot of the interpretation in the text hinges on the comparison between 5-ITu and ZM treatment on the Ndc80 FRET ratios.

We thank the reviewer for pointing out that the connection between Aurora B and tension dependence was not clearly shown in the original manuscript. Agreeing with the point, we replaced endogenous Hec1 with the non-phosphorylatable 9A-Hec1 mutant, and found that the correlation between NDC80 binding and K-K distance is abolished (see Figure 5A in the revised manuscript). This strongly suggests that the phosphorylation of the N-terminal tail of Hec1, which is likely modulated by Aurora B, is responsible for the tension dependency. As the reviewer suggested, we have moved Figure 5—figure supplement 1 of the original manuscript to Figure 5G and 5H.